# DOMAIN GENERALIZATION WITH SMALL DATA

## ABSTRACT

In this work, we propose to tackle the problem of domain generalization in the context of *insufficient samples*. Instead of extracting latent feature embeddings based on deterministic models, we propose to learn a domain-invariant representation based on the probabilistic framework by mapping each data point into probabilistic embeddings. Specifically, we first extend empirical maximum mean discrepancy (MMD) to a novel probabilistic MMD that can measure the discrepancy between mixture distributions (i.e., source domains) consisted of a serial of latent distributions rather than latent points. Moreover, instead of imposing the contrastive semantic alignment (CSA) loss based on pairs of latent points, a novel probabilistic CSA loss encourages positive probabilistic embedding pairs to be closer while pulling other negative ones apart. Benefiting from the learned representation captured by probabilistic models, our proposed method can marriage the measurement on the *distribution over distributions* (i.e., the global perspective alignment) and the distribution-based contrastive semantic alignment (i.e., the local perspective alignment). Extensive experimental results on three challenging medical datasets show the effectiveness of our proposed method in the context of insufficient data compared with state-of-the-art baseline methods.

## 1 INTRODUCTION

Nowadays, we have witnessed a lot of successes via imposing machine learning techniques in a variety of tasks related to computer vision and natural language processing, such as face recognition Li et al. (2022b), object detection Zaidi et al. (2022), and speech recognition Mridha et al. (2022). Despite many achievements so far, the widely-adopted assumption for most existing methods, i.e., it is identically and independently distributed between training and testing data, may not always hold in actual applications Zhou et al. (2022); Liu et al. (2022). In the real-world scenario, it is quite common that the distribution between training and testing data may be different, owing to sophisticated environments. For example, resulting from the differences of device vendor and staining method, acquired histopathological images of breast cancer from different healthcare centers exist significant domain gaps (a.k.a., domain shift, see Figure 1 for more details), which may lead to the catastrophic deterioration of the performance Qi et al. (2020). To address this issue, *domain generalization* (DG) is developed to learn a model from multiple related yet different domains (a.k.a., source domains) that is able to generalize well on unseen testing domain (a.k.a., target domain).

Recently, researchers proposed quite a few domain generalization approaches, such as data augmentation with randomization Yue et al. (2019), data generalization with stylization Verma et al. (2019); Zhou et al. (2021), meta learning Li et al. (2018a); Kim et al. (2021)-based training schemes, among which representation learning-based methods are one of the most popular ones. These representation learning-based methods Balaji et al. (2019) aim to learn domain-invariant feature representation. To be specific, if the discrepancy between source domains in feature space can be minimized, the model is expected to be better generalize well on unseen target domain, owing to learned domain-invariant and transferable feature representation Ben-David et al. (2006). For instance, an classical contrastive semantic alignment (CSA) loss proposed by (Motiian et al., 2017) was to encourage positive sample pairs (with same label) from different domains closer while pulling other negative pairs (with different labels) apart. (Dou et al., 2019) introduced the CSA loss which jointly considers *local class alignment loss* (for point-wise domain alignment) and *global class alignment loss* (for distribution-wise alignment).

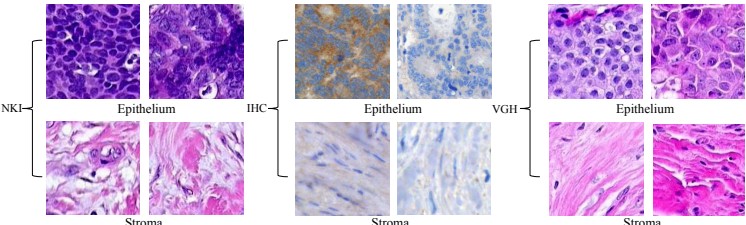

Figure 1: Histopathological image examples of breast cancer tissue from three different healthcare institutes, including NKI with 626 images, IHC with 645 images, and VGH with 1324 images. There are two different tissue types, including epithelium and stroma. Obvious domain gaps (e.g., the density of tissue and the staining color) can be observed.

Despite the progress being achieved so far, it should be noted that a reliable contrastive semantic loss with point-wise (or local) perspective usually requires sufficient samples on source domains such that diverse sample-to-sample pairs can be constructed Sohn (2016); Khosla et al. (2020). For example, (Khosla et al., 2020) proposed a supervised contrastive semantic loss with a considerable volume of batch size on large-scale datasets such that decent performance can be guaranteed. (Yao et al., 2022) also emphasized the importance of the number of sample-to-sample pairs influenced by data sizes for contrastive-based loss on DG problem. Meanwhile, in the eye of distribution-wise (or global) alignment between domains Dou et al. (2019), a consistent distribution measurement (e.g., Kullback–Leibler (KL) divergence) theoretically relies on sufficient samples for the distribution estimation as discussed by (Bu et al., 2018). However, these sufficient samples from multiple source domains may not always be available and accessible, especially for the medical imaging data, due to potential ethic, privacy and proprietorship risks. It is therefore necessary to develop reliable and effective contrastive semantic alignments with local and global perspectives in the context of insufficient samples (a.k.a., small-data scenario) from source domains, in order to achieve better domain-invariant representations.

In this paper, we propose to learn domain-invariant representation from multiple source domains to tackle the domain generalization problem in the context of *insufficient samples*. Instead of extracting latent embeddings (i.e., latent points) based on deterministic models (e.g., convolutional neural networks, CNNs), we propose to leverage a probabilistic framework endowed by variational Bayesian inference to map each data point into probabilistic embeddings (i.e., the latent distribution) for domain generalization. Specifically, by following the domain-invariant learning from global (distribution-wise) perspective, we propose to extend empirical maximum mean discrepancy (MMD) to a novel probabilistic MMD (P-MMD) that can empirically measure the discrepancy between mixture distributions (a.k.a., *distributions over distributions*), consisted of a serial of latent distributions rather than latent points. From a local perspective, instead of imposing the CSA loss based on pairs of latent points, a novel probabilistic contrastive semantic alignment (P-CSA) loss with kernel mean embedding is proposed to encourage positive probabilistic embedding pairs closer while pulling other negative ones apart. Extensive experimental results on three challenging medical imaging classification tasks, including epithelium stroma classification on insufficient histopathological images, imbalanced-class based skin lesion classification, and spinal cord gray matter segmentation, show that our proposed method can achieve better cross-domain performance in the context of insufficient data compared with state-of-the-art baseline methods.

## 2 RELATED WORKS

### 2.1 DOMAIN GENERALIZATION AND ITS APPLICATION IN MEDICAL IMAGE CLASSIFICATION

Existing DG methods can be generally categorized into three different streams, namely data augmentation/generation Yue et al. (2019); Graves (2011); Zhou et al. (2021), meta-learning Li et al. (2018a); Kim et al. (2021) and feature representation learning Li et al. (2018b); Gong et al. (2019); Xiao et al. (2021). Among these methods, feature representation learning, which aims to explore invariant feature information that can be shared across domains, demonstrates to be a widely adopted method for the problem of DG. For feature representation learning-based DG method, Li et al.

(2018b) proposed to conduct multi-domain alignment in latent space via a multi-domain MMD distance. Gong et al. (2019) leveraged adversarial training to eliminate the domain discrepancy such that domain-invariant representation can be learned in a manifold space. Due to the varieties of imaging protocol (e.g., the choice of image solution for MRI image), device vendors (e.g., Philips or Siemens CT scanners), and patient populations (the race and age group), the acquired imaging data from different medical sites may exist significant domain shift problem Liu et al. (2021). Dou et al. (2019) proposed a meta-learning framework to perform local and global semantic alignment for medical image classification. Similar design is also adopted by Li et al. (2022a) for tissue image classification. Qi et al. (2020) utilized the curriculum learning scheme to transfer the knowledge for histopathological images classification. Li et al. (2020a) combined the data augmentation and domain alignment to achieve decent performance on multiple medical data classification tasks. However, these methods may not focus on learning domain-invariant representation on *insufficient samples* from source domains. This scenario may widely encounter in clinical environments because 1) The cost of annotated data by experienced professionals are typically prohibitive, leading to the lack of samples in size and diversity Yoon et al. (2019). 2) For rare diseases (e.g., glioblastoma and lymphoma), the size of data is usually small Lee et al. (2022). 3) medical imaging data are strictly insufficient in most cases due to potential ethic and privacy-preserving concerns Li et al. (2020b).

## 2.2 PROBABILISTIC NEURAL NETWORKS

Compared with deterministic models, probabilistic neural networks turns to learn a distribution over model parameters, which can integrate the uncertainty in predictive modeling Kingma et al. (2015); Gal & Ghahramani (2016). When the data is insufficient, probabilistic models usually can achieve better generalized performance due to its probabilistic property (as an implicit regularization) Blundell et al. (2015). In the context of insufficient samples, Bayesian neural network Neal (2012) (BNN) with variational inference, a representative probabilistic model, not only can improve predictive accuracy as a classifier Wilson & Izmailov (2020), but also can build up the quality of low-dimensional embeddings of insufficient data Mallick et al. (2021), which is a crucial motivation for this paper. Meanwhile, modern analytical approximation techniques (e.g., Variational inference Blei et al. (2017), empirical Bayes Krishnan et al. (2020)) can efficiently infer the posterior distribution of model parameters with stochastic gradient descent method, which can integrate BNN with deterministic DNN conveniently.

In Xiao et al. (2021), the authors proposed to consider the uncertainty of a generalizable model based on BNN, where the distances of positive probabilistic embedding pairs and class distribution are minimized via KL measure. Despite the effectiveness, the dissimilar pairs (i.e., negative pairs) are ignored, which may not benefit feature representation learning. Moreover, they only focused on sample similarity while the distribution information is ignored. Instead, our proposed method comprehensively considers both positive and negative probabilistic embedding pairs via a novel distribution-based contrastive semantic loss.

## 3 METHOD

### 3.1 PRELIMINARY

Assume that there are $K$ domains from different collected environments. The samples in each domain can be represented as $\mathbf{X}_l = \{\mathbf{x}_{l_1}, \cdots, \mathbf{x}_{l_{n_l}}\}$, where $l \in \mathbb{N}^+ : \{1, \cdots, K\}$, $\mathbf{x}_{l_i} \in \mathbb{R}^{d \times 1}$ denotes a sample with the $d$ dimension vector in the $l$-th domain. $n_l$ is the total number of samples in the $l$-th domain. The corresponding labels of samples in each domain can be denoted as $\mathbf{Y}_l = \{\mathbf{y}_{l_1}, \cdots, \mathbf{y}_{l_{n_l}}\}$, where $\mathbf{y}_{l_i} \in \mathbb{R}^{m \times 1}$ is the form of one-hot encoding with $m$ classes in total. For the setting of domain generalization, the source domain data represented as $\{\mathbf{X}_l^S, \mathbf{Y}_l^S\}_{l=1}^K$, can be available in the training phase only, whereas the target domain data, denoted by $\mathbf{X}^T$, are only seen in test phase.

Here, we provide a framework that can learn better domain-invariant representation when there is insufficient source domain data. The probabilistic neural network is imposed to enable high-quality and powerful feature representation in the context of insufficient samples. To effectively perform global perspective alignment, a novel probabilistic MMD is proposed to empirically measure the discrepancy between distributions over distributions based on reproducing kernel Hilbert space. We

also propose a probabilistic contrastive semantic alignment to adapt probabilistic embeddings with local perspective. The details of our proposed method are discussed as below.

## 3.2 PROBABILISTIC EMBEDDING OF INSUFFICIENT DATA

Compared with deterministic models, the probabilistic models can learn a distribution over model weights, which has shown a better capacity to represent latent embeddings Mallick et al. (2021) under insufficient sample scenario, which is a key motivation for this work. Here, Bayesian neural network (BNN) Blei et al. (2017) is utilized to extract the low-dimensional embeddings from high-dimensional inputs. By feeding the inputs into BNN with parameter $\mathbf{W} \sim p(\mathbf{W})$, the samples $\mathbf{X}_l = \{\mathbf{x}_{l_1}, \cdots, \mathbf{x}_{l_{n_l}}\}$ of each domain can be represented by a set of probabilistic embeddings (i.e., latent distributions), i.e., $p(\mathbf{Z}|\mathbf{X}_l) = \{p(\mathbf{z}|\mathbf{x}_{l_1}, \mathbf{W}), \cdots, p(\mathbf{z}|\mathbf{x}_{l_{n_l}}, \mathbf{W})\}$ where $\mathbf{W} \sim p(\mathbf{W})$ is sampled stochastically. The variational inference is used to approximate the posterior distribution of $\mathbf{W}$ with the evidence lower bound (ELBO) (more details can be found in appendix). By using Monte Carlo (MC) estimators with $T$ stochastic sampling operations from the $\mathbf{W}$, the predictive distribution of each $p(\mathbf{z}|\mathbf{x})$ can be unbiased approximation. The number of MC samples and the corresponding issue of computational efficiency is discussed in A.6.

## 3.3 DISTRIBUTION ALIGNMENT VIA PROBABILISTIC MAXIMUM MEAN DISCREPANCY

In this section, we introduce an approach to learning domain-invariant representation from a global perspective by minimizing the discrepancy among domains. Among various distribution distance metrics, Maximum Mean Discrepancy (MMD) is widely adopted Long et al. (2017); Li et al. (2018b) which aims to measure the distance between two probability distributions in a non-parametric manner. Specifically, assume that latent embeddings $\mathbf{Z}_l = \{\mathbf{z}_{l_1}, \cdots, \mathbf{z}_{l_{n_l}}\}$ and $\mathbf{Z}_t = \{\mathbf{z}_{t_1}, \cdots, \mathbf{z}_{t_{n_t}}\}$ are drawn from two unknown distributions $\mathbb{P}_l$ and $\mathbb{P}_t$. The probability measure $\mathbb{P}$ can be mapped into a reproducing kernel Hilbert space (RKHS) $\mathcal{H}$ as a element by setting,

$$\mu_{\mathbb{P}} := \mathbb{E}_{\mathbf{z} \sim \mathbb{P}}[\phi(\mathbf{z})] = \int_{\mathcal{Z}} k(\mathbf{z}, \cdot) d\mathbb{P} = \mathbb{E}_{\mathbf{z} \sim \mathbb{P}}[k(\mathbf{z}, \cdot)], \tag{1}$$

where a reproducing kernel $k : \mathcal{X} \times \mathcal{X} \to \mathbb{R}$ and corresponding feature map $\phi : \mathcal{X} \to \mathcal{H}$ are defined. Let the kernel $k$ is characteristic such that the map $\mu : \mathbb{P} \to \mu_{\mathbb{P}}$ is injective. In this case the MMD can be defined as the distance $\|\mu_{\mathbb{P}_l} - \mu_{\mathbb{P}_k}\|_{\mathcal{H}}$ in $\mathcal{H}$ between mean embeddings and it can be used as a measure of distance between the distributions $\mathbb{P}_l$ and $\mathbb{P}_t$ Borgwardt et al. (2006); Gretton et al. (2012). The explicit computation of MMD can be derived by unbiased empirical estimation of mean map Gretton et al. (2012), i.e.,

$$\mathrm{MMD}\left(\mathbb{P}_l, \mathbb{P}_t\right)^2 = \|\mu_{\mathbb{P}_l} - \mu_{\mathbb{P}_t}\|_{\mathcal{H}}^2 = \left\| \frac{1}{n_l} \sum_{i=1}^{n_l} \phi\left(\mathbf{z}_{l_i}\right) - \frac{1}{n_t} \sum_{j=1}^{n_t} \phi\left(\mathbf{z}_{t_j}\right) \right\|_{\mathcal{H}}^2. \tag{2}$$

The idea of using MMD for domain generalization has been explored in several works (e.g., Li et al. (2018b); Hu et al. (2020)).

In the probabilistic framework, instead of the individual latent embeddings $\mathbf{z}_{l_1,...}$, we have latent probabilistic embeddings $\Pi_{l_1} := p(\mathbf{z}|\mathbf{x}_{l_1}, \mathbf{W}), \ldots$. For a source domain $D_l$, we have the associated *distribution over distributions* $\mathbb{P}_l = \{\Pi_{l_1}, \cdots, \Pi_{l_{n_l}}\}$. For this scenario, we propose to extend the existing *point-based* empirical MMD estimate to a *distribution-based* empirical probability MMD (P-MMD) estimate. P-MMD utilizes empirical estimation by kernels on distributions to measure the discrepancy between mixture distributions $\mathbb{P}_l$ and $\mathbb{P}_t$ under the probabilistic framework.

Specifically, we first represent latent probabilistic embeddings as elements in RKHS $\mathcal{H}_k$ using the kernel $k$, that we call a *level-1* kernel in the sequel, e.g., $\mu_{\Pi l_1} := \mathbb{E}_{\mathbf{z} \sim \Pi_{l_1}}[\phi(\mathbf{z})] = \mathbb{E}_{\mathbf{z} \sim \Pi_{l_1}}[k(\mathbf{z}, \cdot)]$, which is an analogous way to the Eq. (1). The kernel mean embedding $\mu_{\Pi_{l_1}}$ can be regarded as a new feature map for a variety of tasks Yoshikawa et al. (2014). Here, to enable *non-linear* learning on distributions, we introduce *level-2* kernel $K$ Muandet et al. (2012). Consider a level-1 kernel $\kappa$ on $\mathcal{H}$ and its reproducing kernel Hilbert space (RKHS) $\mathcal{H}_\kappa$. Define $K$ as

$$K(\Pi_{l_i}, \Pi_{t_j}) = \kappa(\mu_{\Pi_{l_i}}, \mu_{\Pi_{t_j}}) = \langle \psi(\mu_{\Pi_{l_i}}), \psi(\mu_{\Pi_{t_j}}) \rangle_{\mathcal{H}_\kappa}, \tag{3}$$

where $K$ and its explicit form on kernel mean embeddings $\kappa$ are p.d. kernels Berlinet & Thomas-Agnan (2011). We define the *probabilistic MMD* (P-MMD) empirical estimation method using the *level-2* kernel $K$:

$$
\begin{aligned}
\text{P-MMD}(\mathbb{P}_l, \mathbb{P}_t)^2 &= \|\frac{1}{n_l} \sum_{i=1}^{n_l} \psi(\mu_{\Pi_{l_i}}) - \frac{1}{n_t} \sum_{j=1}^{n_t} \psi(\mu_{\Pi_{t_j}})\|_{\mathcal{H}_\kappa}^2 \\
&= \frac{1}{n_l^2} \sum_{i=1}^{n_l} \sum_{i'=1}^{n_l} K(\Pi_{l_i}, \Pi_{l_{i'}}) + \frac{1}{n_t^2} \sum_{j=1}^{n_t} \sum_{j'=1}^{n_t} K(\Pi_{t_j}, \Pi_{t_{j'}}) \\
&- \frac{2}{n_l n_t} \sum_{i=1}^{n_l} \sum_{j=1}^{n_t} K(\Pi_{l_i}, \Pi_{t_j}).
\end{aligned}
\tag{4}
$$

In this work the level-1 and level-2 kernels, $k$ and $K$, are both Gaussian RBF kernel due to its impressive performance on a limited amount of distribution data Muandet et al. (2012). Namely, K can be represented as

$$
\begin{aligned}
K_{Gau}(\Pi_{l_i}, \Pi_{t_j}) = \kappa(\mu_{\Pi_{l_i}}, \mu_{\Pi_{t_j}}) &= \exp\left(-\frac{\lambda}{2} \|\mu_{\Pi_{l_i}} - \mu_{\Pi_{t_j}}\|_{\mathcal{H}_\kappa}^2\right) \\
&= \exp\left(-\frac{\lambda}{2}(\langle \mu_{\Pi_{l_i}}, \mu_{\Pi_{l_i}} \rangle_{\mathcal{H}_\kappa} - 2\langle \mu_{\Pi_{l_i}}, \mu_{\Pi_{t_j}} \rangle_{\mathcal{H}_\kappa} + \langle \mu_{\Pi_{t_j}}, \mu_{\Pi_{t_j}} \rangle_{\mathcal{H}_\kappa})\right) \\
&= \exp(-\frac{\lambda}{2}(\frac{1}{m_l^2} \sum_{i=1}^{m_l} \sum_{i'=1}^{m_l} k(\mathbf{z}_{l_i}, \mathbf{z}_{l_{i'}}) - \frac{2}{m_l m_t} \sum_{i=1}^{m_l} \sum_{j=1}^{m_t} k(\mathbf{z}_{l_i}, \mathbf{z}_{t_j})) \\
&+ \frac{1}{m_t^2} \sum_{j=1}^{m_t} \sum_{j'=1}^{m_t} k(\mathbf{z}_{t_j}, \mathbf{z}_{t_{j'}}),
\end{aligned}
\tag{5}
$$

where $m_l$ and $m_t$ are determined by sampling times $T$. The kernel mean embedding using the level-1 kernel $k$ creates *distributions* $\mu(\mathbb{P}_1), \ldots, \mu(\mathbb{P}_N)$ represented by the samples $\{\mu_{\Pi_{l_1}}, \ldots, \mu_{\Pi_{l_n}}\}$ for $l = 1, \ldots, N$ respectively in the RKHS $\mathcal{H}_k$. The underlying strategy of P-MMD is to apply the classic MMD to these distributions (with respect to the kernel $\kappa$). To access the effect that the minimization of P-MMD has on the original latent probability distributions across different domains we recall the following:

**Theorem 1** (Muandet et al. (2012)). *Let $\mathbb{P}_1, \ldots, \mathbb{P}_N$ be probability distributions and $\hat{\mathbb{P}} := \frac{1}{N} \sum_{i=1}^N \mathbb{P}_i$. Then the distributional variance given by $\frac{1}{N} \sum \|\mu_{\mathbb{P}i} - \mu_{\hat{\mathbb{P}}}\|$ is 0 iff $\mathbb{P}_1 = \mathbb{P}_2 = \ldots = \mathbb{P}_N$.*

**Corollary 1** (Li et al. (2018b)). *The upper bound of the distributional variance can be written as*

$$
\frac{1}{K^2} \sum_{1 \le i,j \le K} \text{MMD}(\mathbb{P}_i, \mathbb{P}_j)^2.
$$

In our setting Theorem 1 and Corollary 1 along with the fact that $k$ is a characteristic kernel imply the following

**Corollary 2.** $\frac{1}{K^2} \sum_{1 \le i,j \le K} \text{P-MMD}(\mathbb{P}_i, \mathbb{P}_j)^2 = 0$ *iff all moments of latent distributions $\Pi_l$ associated to points of domain $\bar{D}_l$ for $l = 1, \ldots, N$ are distributed identically across domains.*

Following Corollary 2 we define the following loss function:

$$
\mathcal{L}_{global} = \frac{1}{K^2} \sum_{1 \le i,j \le K} \text{P-MMD}(\mathbb{P}_i, \mathbb{P}_j)^2.
\tag{6}
$$

Corollary 2 implies that as 6 tends to 0 so does the distance between the distributions of means, variances and higher moments of the distributions $\Pi_l$ associated to points of different domains.
**Remark.** In the ablation study (see Appendix), we compare the P-MMD approach to simply taking the mean (i.e., first moment) of latent probabilistic embeddings $\Pi_l$ i.e. taking $\Pi_l \to \mathbf{m}_{\Pi_l} = \mathbb{E}_{\mathbf{x} \sim \Pi_l[\mathbf{x}]}$, and then minimizing the associated "vanilla" MMD. Although this scheme is more computational-efficient over our proposed method, it throws away most information about high-level statistics as discussed by (Muandet et al., 2017). We verify empirically that our approach leads to better performance across the domains. The visualized computation of P-MMD is shown in Figure 2.

Although we focus on the scenario of insufficient samples, the computational consumption from Eq. (4) and Eq. (5) may be still prohibitive as the calculation of MMD distance between distributions can

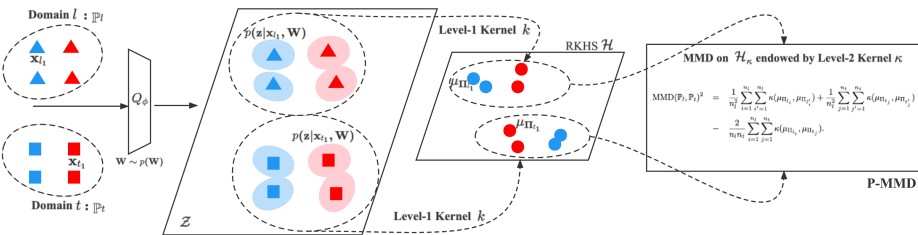

Figure 2: A visualized computational process for probabilistic MMD (P-MMD) on two source domains. The same color for each sample in different domains denotes the same label.

scale at least quadratically with the increasing of sample size (especially for image segmentation task), i.e., $O(n^2)$ in a domain. Here, by following the *linear statistic theory* of MMD, the unbiased estimate can be derived by drawing pairs from two domains with replacement, i.e., P-MMD$(\mathbb{P}_l, \mathbb{P}_t)^2 \approx \frac{2}{n_l} \sum_{i=1}^{\frac{2}{n_l}} \left[ K(\Pi_{l_{2i}}, \Pi_{l'_{2i+1}}) + K(\Pi_{t_{2i}}, \Pi_{t'_{2i+1}}) - K(\Pi_{l_{2i}}, \Pi_{t_{2i+1}}) - K(\Pi_{l_{2i+1}}, \Pi_{t_{2i}}) \right]$, where assuming $n_l = n_t$ for simplicity. (Borgwardt et al., 2006) gives proofs about the unbiased property of the *linear statistic* of MMD and shows that statistic power does not be sacrificed too much.

### 3.4 PROBABILISTIC CONTRASTIVE SEMANTIC ALIGNMENT

To learn domain-invariant representation from local perspective, a popular idea is to encourage positive pairs with same label closer while pulling other negative ones with different labels apart Motiian et al. (2017); Dou et al. (2019). These methods usually measure the Euclidean distance between samples in the embedding space. However, this scheme may not satisfy our probabilistic framework due to its probabilistic embeddings. To this end, we propose a probabilistic contrastive semantic alignment (P-CSA) loss that can utilize the empirical MMD to measure the discrepancy between probabilistic embeddings. The proposed P-CAS loss $\mathcal{L}_{local}$ consists of two components, including the positive probabilistic contrastive loss and negative probabilistic contrastive loss. The former aims to minimize the distance between intra-class distributions from different domains, i.e.,

$$\mathcal{L}_{local}^{pos} = \frac{1}{2} \text{MMD}(\Pi_n, \Pi_q)^2 = \frac{1}{2} \left\| \frac{1}{T} \sum_{i=1}^{T} \phi \left( M_\Theta(\mathbf{z}_{n_i}) \right) - \frac{1}{T} \sum_{j=1}^{T} \phi \left( M_\Theta(\mathbf{z}_{q_j}) \right) \right\|_{\mathcal{H}}^2, s.t. \mathbf{y}_n = \mathbf{y}_q, \quad (7)$$

where $M_\Theta(\cdot)$ denotes the embedding network of metric learning, which will contribute to learn the distance between features better Dou et al. (2019). Then, by introducing a distance margin $\xi$ (can guarantee a appropriate repulsion range), the negative probabilistic contrastive loss is denoted by

$$\begin{aligned} \mathcal{L}_{local}^{neg} &= \frac{1}{2} \max[0, \xi - \text{MMD}(\Pi_n, \Pi_q)^2] \\ &= \frac{1}{2} \max[0, \xi - \left\| \frac{1}{T} \sum_{i=1}^{T} \phi \left( M_\Theta(\mathbf{z}_{n_i}) \right) - \frac{1}{T} \sum_{j=1}^{T} \phi \left( M_\Theta(\mathbf{z}_{q_j}) \right) \right\|_{\mathcal{H}}^2], s.t. \mathbf{y}_n \neq \mathbf{y}_q. \quad (8) \end{aligned}$$

**Model Training.** Our proposed framework consists of three modules, a BNN-based probabilistic extractor $Q_\phi$, a BNN-based classifier $C_\omega$, and a metric network $M_\Theta(\cdot)$. For the $Q_\phi$, we only add a Bayesian layer with ReLU layer on the bottom of a pretrained deterministic model (e.g., ResNet18 by removing fully-connected layers) by following (Xiao et al., 2021). For the $C_\omega$, a Bayesian layer is also introduced to adapt the classification on insufficient sample better. The structure of $M_\Theta$ is the same as (Dou et al., 2019). The images $\mathcal{X} = \{\mathbf{x}_{l_i}\}$ conduct $T$ stochastic forward passes on the $Q_\phi$ and $C_\omega$ by MC sampling to obtain probabilistic predicts $\{\hat{y}_{l_i}^j\}_{j=1}^T$, where the outputs (i.e., probabilistic embeddings) of $Q_\phi$ serve as the inputs for the calculations of $\mathcal{L}_{global}$ and $\mathcal{L}_{local}$. The final predicts $\{\hat{y}_{l_i}^j\}$ are the expectation of $\{\hat{y}_{l_i}^j\}_{j=1}^T$. The total objectives can be summarized as below,

$$\mathcal{L}_{total} = \sum_{l,i} \mathcal{L}_c(\hat{y}_{l_i}, y_{l_i}) + \text{KL}[q_\theta(Q_\phi) \| p(Q_\phi)] + \text{KL}[q_\theta(C_\omega) \| p(C_\omega)] + \beta_1 \mathcal{L}_{local} + \beta_2 \mathcal{L}_{global}, \quad (9)$$

where $\mathcal{L}_c(\hat{y}_{l_i}, y_{l_i})$ is the cross-entropy loss with ground-truth $y_{l_i}$ and its estimation $\hat{y}_{l_i}$. The second and third terms aim to learn a variational distribution $q_\theta(\cdot)$ to approximate the Bayesian posterior distribution on the weights, while minimizing the KL divergence with its prior distribution $p(\cdot)$. The first three terms refer to variational Bayesian inference with ELBO Blei et al. (2017).

Table 1: Domain generalization results on the skin lesion classification task. The average value and standard deviation are reported by running each method with five times. Each column denotes a cross-domain task. For example, in the second column, we use DMF dataset denotes as the target domain and the remaining datasets as the source domains.

| Method | DMF | D7P | MSK | PH2 | SON | UDA | AVG |
|---|---|---|---|---|---|---|---|
| DeepAll | 0.2492 ±0.0127 | 0.5680±0.0181 | 0.6674±0.0083 | 0.8000±0.0167 | 0.8613±0.0296 | 0.6264±0.0312 | 0.6287 |
| MASF Dou et al. (2019) | 0.2692±0.0146 | 0.5678±0.0361 | 0.6815±0.0122 | 0.7833±0.0101 | 0.9204±0.0227 | 0.6538±0.0196 | 0.6460 |
| LDDG Li et al. (2020a) | 0.2793±0.0244 | 0.6007±0.0187 | 0.6967±0.0211 | 0.8167±0.0209 | 0.9272±0.0117 | 0.6978±0.0182 | 0.6697 |
| SWAD Cha et al. (2021) | 0.3582 ±0.0234 | 0.5491 ±0.0231 | 0.6842 ±0.0156 | 0.9167 ±0.0121 | 0.9824 ±0.0012 | 0.7240 ±0.0251 | 0.7024 |
| BDIL Xiao et al. (2021) | 0.2985±0.0452 | **0.6204**±0.0212 | 0.7059±0.0145 | 0.8967±0.0096 | 0.9860±0.0198 | 0.7219±0.0284 | 0.7049 |
| DNA Chu et al. (2022) | 0.3532 ±0.0133 | 0.5581 ±0.0178 | 0.7120 ±0.0194 | 0.9333 ±0.0045 | 0.9851 ±0.0032 | 0.7314 ±0.0141 | 0.7122 |
| DSU Li et al. (2022c) | **0.3830** ±0.0267 | 0.5739 ±0.0147 | 0.6935 ±0.0165 | 0.8833 ±0.0231 | 0.9841 ±0.0098 | 0.7201 ±0.0121 | 0.7063 |
| Ours | 0.3781±0.0136 | 0.6120±0.0115 | **0.7276** ±0.0201 | **0.9416**±0.0103 | **0.9889**±0.0041 | **0.7486** ±0.0123 | **0.7328** |

Table 2: Experiment results of Epithelium Stroma Classification in Histopathological Images. Each column denotes a cross-domain task. For example, in the second column, we use IHC dataset denotes as the target domain and the remaining datasets as the source domains.

| Method | IHC | NKI | VGH | Average (%) |
|---|---|---|---|---|
| DeepAll | 73.29 ±0.13 | 70.60 ±0.15 | 79.56 ±0.11 | 74.48 |
| MASF Dou et al. (2019) | 80.45±0.10 | 76.10±0.11 | 84.44±0.12 | 80.33 |
| SWAD Cha et al. (2021) | 79.74±0.15 | 74.84±0.13 | 84.29±0.12 | 79.62 |
| BDIL Xiao et al. (2021) | 85.56±0.12 | 71.89±0.14 | 85.90±0.18 | 81.05 |
| DNA Chu et al. (2022) | 83.93±0.18 | 73.94±0.15 | 85.57±0.17 | 81.14 |
| DSU Li et al. (2022c) | 81.56±0.14 | 72.47±0.12 | 83.94±0.16 | 79.32 |
| Ours (in this paper) | **88.82**±0.09 | **76.71**±0.10 | **86.92**±0.14 | **84.06** |

## 4 EXPERIMENTS AND ANALYSES

### 4.1 SKIN LESION CLASSIFICATION

Here, we first perform the skin lesion classification task to explore the generalization performance of our proposed method. 7 public skin lesion datasets [1] are utilized, including HAM10000, UDA, SON, DMF, MSK, D7P, and PH2. There are seven classes of skin lesions. The challenges of this task refer to two aspects. 1) ***The diversity of samples:*** The lesion locations (e.g., on the leg and the thigh), patients' characteristics (e.g., the skin age and complexion) and the imaging vendors are different among domains, significant domain shifts thus can not be ignored. 2) ***Insufficient samples:*** The acquired data not only are restricted by the total number of samples in some domains (e.g., UDA and PH2 have only 601 and 200 samples, respectively) but also suffer from the limitation of inter-class imbalance (e.g., SON dataset only has a class of lesion). We follow experimental settings in (Li et al., 2020a), where each dataset is randomly split into 50% training set, 30% test set, and 20% validate set, respectively. The pretrained ResNet18 network is used as the backbone for all methods.

**Results.** Some competitive domain generalization approaches are introduced for comparison, including MASF Dou et al. (2019), BDIL Xiao et al. (2021), LDDG Li et al. (2020a) SWAD Cha et al. (2021), DNA Chu et al. (2022), and DSU Li et al. (2022c). "DeepAll" denotes the model that is trained directly without any domain generalization strategy in all subsequent tasks. We turn their hyperparameters in a wide range. Note that our proposed method, DSU, DNA and BDIL are based on SWAD framework. The accuracy results of each cross-domain task are shown in Table 1. One has some observations as following. First, all methods achieve consistent improvements compared over DeepAll model. Second, we observe that MASF and LDDG with deterministic models may not explicitly adapt to the scenario of insufficient samples. In contrast, our proposed method and other baseline methods impose respective schemes to relieve the impact of insufficient samples, which leads to obvious improvements. Benefiting from probabilistic framework as an implicit regularization, our proposed method and BDIL can learn a distribution over weights, which can handle insufficient samples flexibly. However, it can be observed that additional invariant classifier learning on BDIL may cause negative effects (see the results on DMF) on challenging data, which may be reasonable as the explicit alignment on the classifier with high error probability can lead to negative transfer (take more uncertainty). The lack of explicit domain-invariant representations for DNA and DSU may be difficult to address significant domain shifts compared with our proposed method.

---

[1] https://challenge.isic-archive.com/landing/2018/47/

Table 3: The average results of spinal cord GM segmentation on 4 domain generalization tasks.

| Method | DeepAll | MASF Dou et al. (2019) | LDGG Li et al. (2020a) | KDGG Wang et al. (2021) | DSU Li et al. (2022c) | Ours |
|---|---|---|---|---|---|---|
| DSC ↑ | 0.7425 | 0.7710 | 0.7881 | 0.7886 | 0.7921 | **0.7957** |
| CC ↑ | -11.4 | 23.52 | 34.86 | 33.43 | 34.65 | **35.76** |
| JI ↑ | 0.6160 | 0.6502 | 0.6667 | 0.6667 | 0.6775 | **0.6828** |
| TPR ↑ | 0.7667 | 0.7803 | 0.8058 | 0.8075 | 0.8225 | **0.8260** |
| ASD ↓ | 0.5265 | 0.5505 | 0.4076 | 0.3553 | 0.4362 | **0.3356** |

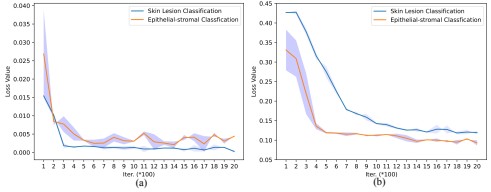

Figure 3: The loss curve of iteration on skin lesion and epothelial-stromal classficaiton tasks. (a) Global alignment loss (b) Local alignment loss.

Table 4: Ablation study on each component of our proposed method for spinal cord gray matter segmentation task (where "site2" is as the target domain). The model on the first row denotes the basic Unet model.

| Backbone (Unet) | Bayesian Layers | Local Alignment | Global Alignment | Bayesian Classifier | DSC | CC | JI | TPR | ASD |
|---|---|---|---|---|---|---|---|---|---|
| ✔ | ✗ | - | - | ✗ | 0.7223 | 26.21 | 0.5789 | 0.8109 | 0.0992 |
| ✔ | ✔ | - | - | ✗ | 0.7934 | 47.19 | 0.6595 | 0.8133 | 0.0692 |
| ✔ | ✔ | - | - | ✔ | 0.8268 | 57.52 | 0.7067 | 0.8156 | 0.0501 |
| ✔ | ✔ | ✔ | ✗ | ✔ | 0.8364 | 60.72 | 0.7195 | 0.8267 | 0.0486 |
| ✔ | ✔ | ✗ | ✔ | ✔ | 0.8371 | 60.57 | 0.7217 | 0.8152 | 0.0510 |
| ✔ | ✔ | ✔ | ✔ | ✔ | **0.8485** | **63.78** | **0.7389** | **0.8401** | **0.0401** |

## 4.2 EPITHELIUM STROMA CLASSIFICATION

The epithelium-stroma ratio can reflect the prognostic status of the tumor, especially for the breast cancer. A key step, therefore, is to recognize which tissues are epithelial or stromal in histopathological images. The obvious domain gaps can be observed from Figure 1. Meanwhile, it is much difficult to collect massive number of histopathological images from different sites due to the privacy. Here, three histopathological image datasets [2] collected from different medical institutes are used for comparison, where the NKI and VGH datasets only have 671 and 615 images, respectively. We follow the research in Qi et al. (2020) to extract epithelial or stromal patches from histopathological images, in order to balance the number of images among datasets. Then, IHC, NKI, and VGH datasets have 1342, 1230, and 1376 patches, respectively, which is still insufficient for training. We utilize the DomainBed benchmark Gulrajani & Lopez-Paz (2020) for fair comparison, where each dataset in source domain is randomly split into 80% training set, 20% validate set. The testing is on overall target domain. The pretrained ResNet18 is adopted by all methods as backbone.

**Results.** We compare our proposed method with recent DG models, including MASF, SWAD, BDIL, DNA, and DUS. By turning the hyperparameter of baseline methods in a wide range, the classification accuracy on each target domain (the remaining is as the source domain) is reported in Table 2. Some observations can be summarized as following. First, we observe that the type of weight averaging method (e.g., SWAD and DNA) is effective for this challenging out-of-domain task. However, due to the lack of explicit domain alignment, the obvious domain shifts may not be fully addressed via weighted ensemble learning, leading to the limitation of the performance. Second, BDIL not only adopts two-level alignments on feature extractor and classifier, but also obtains further improvements by probabilistic framework. However, one can observe that BDIL has a similar performance drop (in the skin lesion classification) on the challenging task (i.e., the NKI task). The best performance achieved by our proposed method thus shows the effectiveness of our proposed method. DSU has the poorest performance among all baseline methods, which may be reasonable as the straightforward domain randomization in feature space may not be powerful for eliminating obvious domain shifts.

## 4.3 SPINAL CORD GRAY MATTER SEGMENTATION

The spinal cord Gray Matter (GM) segmentation Challenge Dataset [3] is used here, where the acquired magnetic resonance imaging (MRI) data are collected from four healthcare centers, and acquisition manufactures and imaging protocols are variable. The challenges of insufficient sample are from two aspects. 1) The number of slices in some sites is relatively small (e.g., site1 and site2 have only 30 and 113 slices). 2) The number of target pixels is small, as GM area is only a very small area in overall slice. We follow the training protocols used in Li et al. (2020a) for all methods.

---

[2]http://fimm.webmicroscope.net/supplements/epistroma
[3]http://niftyweb.cs.ucl.ac.uk/challenge/index.php

Table 5: Experiment results of PACS multi-domain classification task based on ResNet50. Each column denotes a cross-domain task. For example, in the third column, we use Art dataset denotes as the target domain and the remaining datasets as the source domains.

| Method | Reference | Art | Cartoon | Photo | Sketch | Average (%) |
|---|---|---|---|---|---|---|
| RSC Huang et al. (2020) | ECCV 2020 | 78.9 | 76.9 | 94.1 | 76.8 | 81.7 |
| L2A-OT Zhou et al. (2020) | ECCV 2020 | 83.3 | 78.2 | 96.2 | 73.6 | 82.8 |
| MatchDG Mahajan et al. (2021) | ICML 2020 | 81.2 | 80.4 | 96.8 | 77.2 | 83.9 |
| pAdaIN Nuriel et al. (2021) | CVPR 2021 | 81.7 | 76.6 | 96.3 | 75.1 | 82.5 |
| MixStyle Zhou et al. (2021) | ICLR 2021 | 86.8 | 79.0 | 96.6 | 78.5 | 85.2 |
| SagNet Nam et al. (2021) | CVPR 2021 | 87.4 | 80.7 | 97.1 | 80.0 | 86.3 |
| SWAD Cha et al. (2021) | NeurIPS 2021 | 89.3 | 83.4 | 97.3 | 82.5 | 88.1 |
| DNA Chu et al. (2022) | ICML 2022 | 89.8 | 83.4 | 97.7 | 82.6 | 88.4 |
| **Bayesian** | - | 89.4 | 83.5 | 97.3 | 82.3 | 88.1 |
| **Ours** | - | **90.2** | **85.2** | **98.7** | **83.6** | **89.4** |

**Results.** Here, four domain generalization approaches are utilized for comparison, including MASF, LDDG Li et al. (2020a), KDDG Wang et al. (2021), and DSU. To qualitatively evaluate the segmentation results, 5 complementary metrics are introduced from statistical and distance-based perspectives, respectively. The average results on four domain generalization tasks are illustrated in Table 3. The detailed evaluation results for each domain can be found in Appendix. First, the performance of segmentation results among all methods achieve improvements with an obvious margin compared with DeepAll. Second, suffering from insufficient samples in some domains, LDDG and KDDG with deterministic models may not model these uncertainties explicitly. In contrast, our proposed method and DSU can generally obtained the best and second best results.

## 4.4 ADDITIONAL RESULTS

**Ablation study on each component of our proposed method.** We are interested in the effectiveness of each component of our proposed method. The results can be shown in Table 4. First, we observe that better performance can be achieved by introducing probabilistic layer compared with the results that using Unet, which reflects the superiority of probabilistic models. Secondly, we observe that by either introducing local or global alignment for domain-invariant information learning, better performance can be achieved compared with the results of only using probabilistic layer, which shows the effectiveness of introduced probabilistic feature regularization term. Last but not the least, by imposing the domain-invariant learning with both local and global views, the performances are further improved, which justifies the effectiveness of our proposed method by jointly considering local and global alignment.

**Effectiveness of domain-invariant loss.** We are also interested in impacts of domain-invariant losses on different tasks. The results can be shown in Figure 3. As we can observe, for the skin lesion (on DMF) and epithelium-stroma (on IHC) classification tasks, the loss curves with iterations reflect the global discrepancy converges faster than local discrepancy, while the more challenging cross-domain task converges more slowly on global alignment.

**Results on DG Benchmark.** While our proposed method is designed for the context of insufficient data, it can also be applied to the setting of conventional DG problem generalization. Here, we introduce three DG benchmarks, namely PACS, OfficeHome and VLCS, for further comparison. Compared with some large-scale benchmarks (e.g., DomainNet and Wilds), these two datasets are more appropriate to explore the effectiveness of different DG models under the scenario of insufficient samples. We report the results on PACS in Table 5. We compare our proposed method with some state-of-the-art DG methods. To be fair, all methods adopt a same backbone, i.e., the pretrained ResNet50. "Bayesian" model does not have any alignment compared with our model. As we can see, our proposed method outperforms recent methods, such as DNA and SWAD. Specifically, our proposed method surpasses the gradient operation-based method (e.g., RSC). Although data generation methods (e.g., MixStyle) can effectively tackle the insufficient sample problem via additional generative samples, the lack of effective domain-invariant learning may hamper the improvement of the performance.

## 5 CONCLUSION

In this work, we address the domain generalization problem in the context of insufficient data from source domains. Benefiting from the learned representation captured by probabilistic models, our proposed method can marriage the measurement on the *distribution over distributions* by level-2 kernel and probabilistic contrastive semantic alignment. Extensive experiments on challenging medical image tasks indicate the effectiveness of our proposed method.

## ETHICS STATEMENT

We believe that there is no ethics issue in our work. The reasons are provided as follow. First, no personal or private information exists in our adopted medical imaging data. Second, access to these medical imaging data is feasible by signing an agreement form with the provider or download datasets directly from our given website link in the main content. Third, in our submission, we focus on the problem of domain generalization instead of long-tail/imbalance data classification. We therefore follow the previous work on these medical imaging datasets.

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

## A  APPENDIX

### A.1  DETAILS OF BAYESIAN NEURAL NETWORK

For our proposed method, the Bayesian layer refers to the probabilistic extractor $Q_\phi$ and the probabilistic classifier $C_\omega$. Here, a simple and convenient PyTorch library, namely BayesianTorch Krishnan et al. (2022), is utilized to construct the Bayesian neural network. The log evidence lower bound (ELBO) cost function, i.e.,

$$\mathcal{L} := \int q_\theta log(y|x, w) dw - \text{KL}[q_\theta(w)|p(w)], \tag{10}$$

can be calculated automatically. By using BayesianTorch, arbitrary deterministic models can be converted into the Bayesian layers easily. In this paper, mean-field variational inference (MFVI) Graves (2011) is adopted, where the parameters of the model are characterized by fully factorized Gaussian distribution endowed by variational parameters $\mu$ and $\sigma$, i.e.,

$$q_\theta(w) := \mathcal{N}(w|\mu, \sigma). \tag{11}$$

By using stochastic gradient descent method with ELBO cost, the variational distribution $q_\theta(w)$ as the approximation of the posterior distribution, and corresponding parameters ($\mu$ and $\sigma$) and can be learned conveniently.

For the settings of Bayesian layer, we follow the model priors with empirical Bayes using DNN (MOPED) method for the parameter settings of weights prior, each weight is sampled from the Gaussian distribution independently Krishnan et al. (2020),

$$w \sim \mathcal{N}(w_{\text{DNN}}, \delta|w_{\text{DNN}}|), \tag{12}$$

where $w_{\text{DNN}}$ denotes the mean of prior distribution from the maximum likelihood estimates of weights from deterministic deep neural network. $\delta$, a hyperparameter, is set to the initial perturbation factor for the percentage of the pretrained deterministic weight values. The variational layer is modeled using reparameterization trick. The MOPED can realize better training convergence for complex models Krishnan et al. (2020), which is beneficial to our proposed method. In this paper, we follow the setting in (Krishnan et al., 2020) to set the initial perturbation factor $\delta$ for the weight to $0.1$.

### A.2  EXPERIMENTAL DETAILS OF SKIN LESION CLASSIFICATION

**Dataset Details.** There are seven classes of skin lesions, including melanoma (*mel*), melanocytic nevus (*nv*), dermato broma (*df*), basal cell carcinoma (*bcc*) benign keratosis (*bkl*), vascular lesion (*vasc*), and actinic keratosis (*akiec*). The 7 public skin lesion datasets suffer from an insufficient data problem from some (certain) source domains. For example, the PH2 and UDA datasets only have 200 and 601 skin lesion images, respectively. The number of images for each domain can be found in Table 9. More details of datasets can be found in Yoon et al. (2019). For inputs, all images are resized into $224 \times 224$ for all methods.

**Implementation Details.** The pretrained ResNet18 is introduced as the backbone for all methods. For our proposed method, the structure of Bayesian layer in probabilistic extractor $Q_\phi$ is a fully-connected based Bayesian neural network with $512 \times 512$. Note that the DSU, BDIL, DNA, and our proposed method are constructed based on SWAD framework. The hyperparameters of SWAD follow default settings Cha et al. (2021). The DSU can be regarded as the uncertainty version of SWAD with ResNet18. The structure of Bayesian layer in probabilistic classifier $C_\omega$ is also a fully-connected based Bayesian neural network with $512 \times 7$. The construction of $C_\omega$ is the same as that of $T_\phi$. Due to the class imbalance problem, the focal loss Lin et al. (2017) as the classification objective is introduced for all methods.

During training, our proposed method is optimized by Adam optimizer with $5 \times 10^{-5}$ learning rate. The training steps are 2000. For each step, we randomly sample from each training source domain with 32 samples to construct the mini-batch. To evaluate the testing set, the training process is stopped according to the validation loss computed by SWAD on the validation set. The hyperparameters are also selected in a wide range on the validation set. For the probabilistic MMD, level-1 and leve-2 kernels are the Gaussian RBF kernels by following (Muandet et al., 2012). The kernel bandwidth is empirically set to 1 for all kernels. For the probabilistic CSA loss, the distance margin $\xi$ is set

Table 6: Domain generalization results on gray matter segmentation task. For the DSC, CC, TPR, and JI, the higher the better. For the ASD, the lower the better.

(a) DeepAll

| source | target | DSC | CC | JI | TPR | ASD |
|---|---|---|---|---|---|---|
| 2,3,4 | 1 | 0.8560 | 65.34 | 0.7520 | 0.8746 | 0.0809 |
| 1,3,4 | 2 | 0.7323 | 26.21 | 0.5789 | 0.8109 | 0.0992 |
| 1,2,4 | 3 | 0.5041 | -209 | 0.3504 | 0.4926 | 1.8661 |
| 1,2,3 | 4 | 0.8775 | 71.92 | 0.7827 | 0.8888 | 0.0599 |
| Average | | 0.7425 | -11.4 | 0.6160 | 0.7667 | 0.5265 |

(b) KDDG

| source | target | DSC | CC | JI | TPR | ASD |
|---|---|---|---|---|---|---|
| 2,3,4 | 1 | 0.8745 | 70.75 | 0.7795 | 0.8949 | 0.0539 |
| 1,3,4 | 2 | 0.8229 | 56.71 | 0.6997 | 0.8226 | 0.0490 |
| 1,2,4 | 3 | **0.5676** | -63.1 | 0.3866 | 0.5904 | 1.2805 |
| 1,2,3 | 4 | 0.8894 | 75.06 | 0.8011 | 0.9222 | 0.0377 |
| Average | | 0.7886 | 34.86 | 0.6667 | 0.8075 | 0.3553 |

(c) MASF

| source | target | DSC | CC | JI | TPR | ASD |
|---|---|---|---|---|---|---|
| 2,3,4 | 1 | 0.8502 | 64.22 | 0.7415 | 0.8903 | 0.2274 |
| 1,3,4 | 2 | 0.8115 | 53.04 | 0.6844 | 0.8161 | 0.0826 |
| 1,2,4 | 3 | 0.5285 | -99.3 | 0.3665 | 0.5155 | 1.8554 |
| 1,2,3 | 4 | **0.8938** | **76.14** | **0.8083** | 0.8991 | 0.0366 |
| Average | | 0.7710 | 23.52 | 0.6502 | 0.7803 | 0.5505 |

(d) LDDG

| source | target | DSC | CC | JI | TPR | ASD |
|---|---|---|---|---|---|---|
| 2,3,4 | 1 | 0.8708 | 69.29 | 0.7753 | 0.8978 | **0.0411** |
| 1,3,4 | 2 | 0.8364 | 60.58 | 0.7199 | 0.8485 | 0.0416 |
| 1,2,4 | 3 | 0.5543 | -71.6 | 0.3889 | 0.5923 | 1.5187 |
| 1,2,3 | 4 | 0.8910 | 75.46 | 0.8039 | 0.8844 | **0.0289** |
| Average | | 0.7881 | 33.43 | 0.6720 | 0.8058 | 0.4076 |

(e) DSU

| source | target | DSC | CC | JI | TPR | ASD |
|---|---|---|---|---|---|---|
| 2,3,4 | 1 | 0.8739 | 70.32 | 0.7794 | 0.9210 | 0.0793 |
| 1,3,4 | 2 | 0.8474 | 63.58 | 0.7367 | **0.8502** | 0.0494 |
| 1,2,4 | 3 | 0.5574 | -70.4 | 0.3923 | 0.6097 | 1.5049 |
| 1,2,3 | 4 | 0.8897 | 75.10 | 0.8018 | 0.9225 | 0.0415 |
| Average | | 0.7921 | 34.65 | 0.6775 | 0.8225 | 0.4362 |

(f) Ours

| source | target | DSC | CC | JI | TPR | ASD |
|---|---|---|---|---|---|---|
| 2,3,4 | 1 | **0.8786** | **71.57** | **0.7873** | **0.9293** | 0.0422 |
| 1,3,4 | 2 | **0.8485** | **63.78** | **0.7389** | 0.8401 | **0.0401** |
| 1,2,4 | 3 | 0.5634 | -68.0 | **0.3992** | **0.6103** | 1.2239 |
| 1,2,3 | 4 | 0.8921 | 75.69 | 0.8058 | **0.9245** | 0.0362 |
| Average | | **0.7957** | **35.76** | **0.6828** | **0.8260** | **0.3356** |

to 1. For the $\mathcal{L}_{local}$ and the $\mathcal{L}_{global}$, the $\beta_1$ and $\beta_2$ are 0.1 and 0.7, respectively. By balancing the performance and computational efficiency, $T$, the number of Monte Carlo sampling in each Bayesian layer, is 10.

## A.3 EXPERIMENTAL DETAILS OF EPITHELIUM STROMA CLASSIFICATION

**Dataset Details.** There are two types of basic tissues, i.e., the epithelium and the stroma. Due to the differences of the scanner, the staining type, and the population, the color of the background and the morphological structure among different histopathological image datasets are diverse. The number of images for each domain can be found in Table 9. The extract epithelial or stromal patches are resized into $224 \times 224$.

**Implementation Details.** The pretrained ResNet18 is utilized as the backbone for all methods. The basic model framework and corresponding parameters for our proposed method are similar with the settings mentioned in A.2. The DSU, BDIL, DNA, and our proposed method are constructed based on SWAD framework with DomainBed benchmark. The classification objective is the cross-entropy loss with softmax function.

During training, our proposed method is optimized by Adam optimizer with $5 \times 10^{-5}$ learning rate. The training steps are 4000. The holdout fraction rate for DomainBed is set to 0.2 for all methods such that the hyperparameters can be selected in a wide range on the validation set. The $\beta_1$ and $\beta_2$ are 0.1 and 0.7 for the $\mathcal{L}_{local}$ and the $\mathcal{L}_{global}$, respectively. Other hyperparameters are the same as the settings mentioned in A.2.

## A.4 EXPERIMENTAL DETAILS OF SPINAL CORD GRAY MATTER SEGMENTATION

**Dataset Details.** The spinal cord gray matter (GM) segmentation is an emergent task that can be utilized to predict disability (as a biomarker) via evaluating the atrophy of GM area. The acquired magnetic resonance imaging (MRI) data are collected from four healthcare centers (including "site1", "site2","site3", and "site4"), where acquisition manufacturers (including Philips Achieva, Siemens Trio, and Siemens Skyra) and imaging protocols (lead to the difference in the resolution of the voxel) are variable. The number of images for each domain can be found in Table 9. By following (Li et al.,

Table 7: **Out-of-domain accuracies (%) on** OfficeHome based on ResNet50.

| Algorithm | Art | Clipart | Product | Real | Avg |
|---|---|---|---|---|---|
| Mixstyle Zhou et al. (2021) | 51.1 | 53.2 | 68.2 | 69.2 | 60.4 |
| RSC Huang et al. (2020) | 60.7 | 51.4 | 74.8 | 75.1 | 65.5 |
| DANN Ganin et al. (2016) | 59.9 | 53.0 | 73.6 | 76.9 | 65.9 |
| GroupDRO Sagawa et al. (2019) | 60.4 | 52.7 | 75.0 | 76.0 | 66.0 |
| MTL Blanchard et al. (2021) | 61.5 | 52.4 | 74.9 | 76.8 | 66.4 |
| VREx Krueger et al. (2021) | 60.7 | 53.0 | 75.3 | 76.6 | 66.4 |
| MLDG Balaji et al. (2018) | 61.5 | 53.2 | 75.0 | 77.5 | 66.8 |
| SagNet Qian et al. (2021) | 63.4 | 54.8 | 75.8 | 78.3 | 68.1 |
| CORAL Sun & Saenko (2016) | 65.3 | 54.4 | 76.5 | 78.4 | 68.7 |
| SWAD Cha et al. (2021) | 66.1 | 57.7 | 78.4 | 80.2 | 70.6 |
| DNA Chu et al. (2022) | 67.7 | 57.7 | 78.9 | 80.5 | 71.2 |
| **Bayesian** | 67.0 | 58.0 | 79.3 | 80.4 | 71.2 |
| **Ours** | **68.2** | 58.9 | **80.2** | 80.7 | **72.0** |

Table 8: **Out-of-domain accuracies (%) on** VLCS based on ResNet50.

| Algorithm | C | L | S | V | Avg |
|---|---|---|---|---|---|
| Mixstyle Zhou et al. (2021) | 98.3 | 64.8 | 72.1 | 74.3 | 77.4 |
| RSC Huang et al. (2020) | 97.9 | 62.5 | 72.3 | 75.6 | 77.1 |
| DANN Ganin et al. (2016) | **99.0** | 65.1 | 73.1 | 77.2 | 78.6 |
| GroupDRO Sagawa et al. (2019) | 97.3 | 63.4 | 69.5 | 76.7 | 76.7 |
| MTL Blanchard et al. (2021) | 97.8 | 64.3 | 71.5 | 75.3 | 77.2 |
| VREx Krueger et al. (2021) | 98.4 | 64.4 | 74.1 | 76.2 | 78.3 |
| MLDG Balaji et al. (2018) | 97.4 | 65.2 | 71.0 | 75.3 | 77.2 |
| SagNet Qian et al. (2021) | 97.9 | 64.5 | 71.4 | 77.5 | 77.8 |
| CORAL Sun & Saenko (2016) | 98.3 | **66.1** | 73.4 | 77.5 | 78.8 |
| SWAD Cha et al. (2021) | 98.8 | 63.3 | 75.3 | 79.2 | 79.1 |
| DNA Chu et al. (2022) | 98.8 | 63.6 | 74.1 | 79.5 | 79.0 |
| **Ours** | 98.9 | 63.4 | **75.8** | **79.8** | **79.5** |

2018b), the 3D MRI data are split into 2D slices in axial view. Then, these obtained 2D slices are centered cropped to $160 \times 160$ and randomly cropped to $144 \times 144$ for training.

**Implementation Details.** The 2D-Unet Ronneberger et al. (2015) is leveraged as the backbone for all methods. For our proposed method, probabilistic extractor $Q_\phi$ is constructed by two Bayesian-based $1 \times 1$ convolutional layers. The input and output channels in the first convolutional layer are both 64. After a ReLU layer, the input and output channels in the second convolutional laye are 64 and 1, respectively. The BayesianTorch can enable to convert ordinary convolutional layer into Bayesian convolutional neural network easily. The Bayesian neural network adopts MFVI to approximate the posterior distribution of weights. The parameters of Bayesian layer are the same as aforementioned settings. The structure of Bayesian layer in probabilistic classifier $C_\omega$ is a Bayesian-based $1 \times 1$ convolutional layers. The input and output channels are 64 and 1, respectively. The construction of $C_\omega$ is the same as that of $T_\phi$. Here, all methods adopt a two-stage scheme for coarse-to-fine segmentation, as used in (Li et al., 2020a). Specifically, we first conduct preliminary segmentation to obtain the spinal cord area from the original 2D slice. Then, we perform elaborative segmentation on obtained spinal cord results to derive gray matter results.

Here, the settings of most hyperparameters follow (Li et al., 2020a), where the Adam optimizer is utilized with learning rate as $1 \times 10^{-4}$, weight decay as $1 \times 10^{-8}$. We randomly select 8 slices from each source domain to construct the mini-batch. All models are trained with 200 epochs, where the learning rate will be decreased each 80 epochs with a factor of 10. Other hyperparameters such as kernel function, kernel bandwidth and distance margin are similar with the settings in skin lesion classification and epithelium-stroma classification. The segmentation can be regarded as the pixel-level classification. For the $\mathcal{L}_{local}$ and $\mathcal{L}_{global}$, we follow (Motiian et al., 2017) to randomly

Table 9: The details of adopted datasets

| Task | Datasets (Domains) and Corresponding Size | Number of Class |
|---|---|---|
| Skin Lesion Classification | HAM:10015; DMF:1212; D7P:1926;MSK:3551; UDA:601; PH2:200; SON:9251 | 7 |
| Epithelium Stroma Classification | NKI: 671; IHC:1376; VGH: 615 | 2 |
| Spnal Cord GM Segementation | site1: 30; site2: 113; site3: 246; site4: 122 | 2 (pixel-level) |
| PACS | Art: 2048; Cartoon:2344; Photo:1670; Sketch:3929 | 7 |
| OfficeHome | A total of around 15500 images for 4 domains (Art, Clipart, Product, and Real with around 3897 per domain) | 65 |
| VLCS | VOC2007 (V): 3376; LabelMe (L): 2656; SUN09(S):3282; Caltech101 (C): 1415 | 5 |

sample some positive and negative pairs from two domains such that the computational efficiency can be improved significantly. Here, we randomly sample 400 positive and negative pixel pairs from two domains in a mini-batch for the computation of $\mathcal{L}_{local}$, respective. By leveraging selected pixels of a domain in $\mathcal{L}_{local}$, we further utilize these pixels to calculate the $\mathcal{L}_{global}$, which may induce a more accurate measurement owing to the balanced class distribution, as well as reducing computational cost. For the $\mathcal{L}_{local}$ and $\mathcal{L}_{global}$, the $\beta_1$ and the $\beta_2$ are set to 0.01 and 0.001.

**Result Analysis.** Dice Similarity Coefficient (DSC), Jaccard Index (JI), and Conformity Coefficient (CC) are used to measure the accuracy of obtained segmentation results. Besides, True Positive Rate (TPR) and Average Surface Distance (ASD) are introduced as complementary evaluations from statistical and distance-based perspectives. The experimental results are shown in Table 6 in details. As we can see, our proposed method can achieve best or second-best performance in all task. For average results, our proposed method and DSU roughly achieve the best and second-best performance, especially in the DSC, JI, and TPR, which may be reasonable. Specifically, DSU introduce the multivariate Gaussian distribution of feature statistics for the uncertainty of the feature. Our proposed method not only can model the uncertainty by the introduction of Bayesian neural network, but also can learn distribution-based domain-invariant representations in latent feature space.

## A.5 EXPERIMENTAL SETTINGS AND ADDITIONAL RESULTS ON BENCHMARKS

Besides PACS benchmark dataset, we further validate the effectiveness of our proposed method on two popular benchmark datasets, including OfficeHome (has 15588 samples with 65 classes from four domains) and VLCS (has 10729 samples with 5 classes from four domains). The number of images for each domain can be found in Table 9. We adopt pretrained ResNet50 as the backbone for all benchmarks. The structure of the overall framework is similar with the model mentioned in lesion skin classification. Our proposed method as well as baseline methods are all based on DomainBed, where the holdout fraction (the proportion of validation set) rate for DomainBed is set to 0.2 for all methods. A domain is the target domain and the remaining domains are the source domain for training. The testing is on the overall data of a target domain.

Here, our proposed method is optimized by Adam optimizer with learning rate as $5 \times 10^{-5}$. The batch size for each source domain is 32. The training steps are set to 20000 for PACS and OfficeHome, and 2000 for VLCS. By following the SWAD framework, the training process will be stopped for our proposed method when the validation loss increases significantly. The hyperparameters are selected in a wide range on the validation set. For the $\mathcal{L}_{local}$ and $\mathcal{L}_{global}$, the $\beta_1$ and the $\beta_2$ are set to 0.1 and 1 for all benchmark datasets. Other hyperparameters such as kernel function, kernel bandwidth and distance margin are similar with the settings mentioned before.

The experimental results on OfficeHome and VLCS can be shown in Table 7 and Table 8. As we can see, our proposed method achieves better performance compared with state-of-the-art methods, such as SWAD and DNA. Compared with domain-invariant based approaches (e.g., DANN), our proposed method has a significant improvement due to the introduction of probabilistic framework. Meanwhile, the model, namely "Bayesian" on Table 7 can be regarded as a probabilistic version of

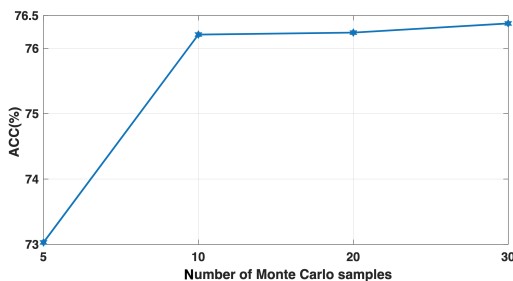

Figure 4: The performance of our proposed model on the NKI task of Epithelium Stroma classification with different Monte Carlo samples $T$.

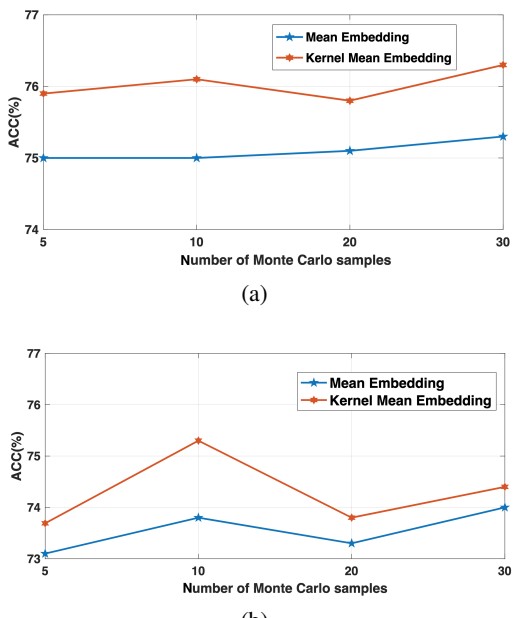

Figure 5: The performance comparison between mean embedding method and kernel mean embedding method with different Monte Carlo samples $T$. For each sub-figure, we use only one alignment operation. (a) Local alignment. **Mean Embedding:** The mean embedding operation with Euclidean distance is utilized between probabilistic embedding pairs. **Kernel Mean Embedding:** The kernel mean embedding with MMD distance is utilized between probabilistic embedding pairs. (b) Global alignment. **Mean Embedding:** The mean embedding operation with MMD distance is utilized between domains (as distributions). **Kernel Mean Embedding:** The kernel mean embedding with P-MMD distance is utilized between domains (as distributions over distributions).

SWAD via replacing deterministic layers with Bayesian layers. Interestingly, compared with SWAD, a significant improvement can be obtained by Bayesian model (which does not have any alignment compared with our model), which shows the effectiveness of probabilistic framework on insufficient data. Our proposed method also outperforms the data augmentation-based approach (e.g., Mixstyle).

## A.6 ADDITIONAL ANALYSIS

First, it is much important to balance the number of Monte Carlos samples and the computational efficiency. On the one hand, the property of probabilistic embeddings can be affected by the Monte Carlos sampling. On the other hand, too many Monte Carlos samples may suffer from the heavy computational cost. (Xiao et al., 2021) suggested that the distributional property and computational cost are both acceptable for the computation of the KL divergence when the number of Monte Carlos

samples is chosen appropriately, the practical performance for our proposed method need to be explored. We conduct the experiments on the NKI task of Epithelium-Stromal classification with different Monte Carlo samples $T$.

The results are shown in Figure 4. As we can see, if the number of Monte Carlo samples is too small, it is difficult to capture the property of distribution for probabilistic embeddings. As the increase of $T$, there is an obvious improvement for our proposed method. Interestingly, the performance is gradually saturated. As a result, by balancing the number of Monte Carlos samples and the computational efficiency, the number of Monte Carlos samples $T$ in each Bayesian layer is set to 10.

**Kernel Mean Embedding (level-2 kernel)** *v.s.* **Mean Embedding.** Second, we explore the effect of different schemes for probabilistic embeddings. A straightforward method is first to represent probabilistic embeddings with the expectation (i.e., first moment), which is called as the *Mean Embedding*. Then, a probabilistic embedding can be regarded as a latent point, and the MMD can be leveraged to measure the discrepancy between distributions consisted of latent points.

For the mean embedding-based $\mathcal{L}_{global}$, the computational process of this scheme for MMD distance can be formulated as

$$\text{MMD}(\mathbb{P}_l, \mathbb{P}_t)^2 = \| \frac{1}{n_l} \sum_{i=1}^{n_l} \varphi(\mathbb{E}[\Pi_{l_i}]) - \frac{1}{n_t} \sum_{j=1}^{n_t} \varphi(\mathbb{E}[\Pi_{t_j}]) \|_{\mathcal{H}}^2. \tag{13}$$

The Eq. (13) can be further constructed a global alignment loss $\mathcal{L}_{global}$. For the local alignment loss $\mathcal{L}_{local}$, the Euclidean distance can be used to compute the distance between latent points, which is similar with original CAS loss in (Motiian et al., 2017). For the positive pairs with the same label, the mean embedding-based positive contrastive loss $\mathcal{L}_{local}^{pos}$ can be represented as

$$\mathcal{L}_{local}^{pos} = \frac{1}{2} \left\| \frac{1}{T} \sum_{i=1}^{T} \mathbb{E}\left[M_{\Theta}(\mathbf{z}_{n_i})\right] - \frac{1}{T} \sum_{j=1}^{T} \mathbb{E}\left[M_{\Theta}(\mathbf{z}_{q_j})\right] \right\|_2^2, \, s.t. \mathbf{y}_n = \mathbf{y}_q, \tag{14}$$

where $M_{\Theta}(\cdot)$ denotes the embedding network of metric learning. For the negative pairs with the different labels, the negative contrastive loss is denoted by

$$\mathcal{L}_{local}^{neg} = \frac{1}{2} \max[0, \xi - \left\| \frac{1}{T} \sum_{i=1}^{T} \mathbb{E}\left[M_{\Theta}(\mathbf{z}_{n_i})\right] - \frac{1}{T} \sum_{j=1}^{T} \mathbb{E}\left[M_{\Theta}(\mathbf{z}_{q_j})\right] \right\|_2^2], s.t. \mathbf{y}_n \neq \mathbf{y}_q. \tag{15}$$

As a result, a mean embedding-based contrastive loss with the view of local alignment can be calculated as

$$\mathcal{L}_{local} = \mathcal{L}_{local}^{pos} + \mathcal{L}_{local}^{neg}. \tag{16}$$

Instead, we can observe from Figure 2 that our proposed method induces a level-2 kernel-based MMD with empirical estimation for probabilistic embeddings. Specifically, our proposed scheme can preserve higher moments of a probabilistic embedding via nonlinear level-1 kernel (see the fourth component in Figure 2). Moreover, by introducing a level-2 kernel, the similarities between probabilistic embeddings also can be measured based on their own moment information (see the last component in Figure 2). Benefiting from these virtues, the proposed probabilistic MMD can accurately capture the discrepancy between mixture distributions via an extended empirical MMD fashion.

Here, we validate the effectiveness of different schemes on the NKI task of Epithelium Stroma classification in each aligned view. The experimental settings are similar for different methods. The experimental results can be found in Figure 5. As we can see, our proposed method achieves consistent improvements in each alignment method with different Monte Carlo samples, which may be reasonable as the kernel mean representation can preserve many statistical components due to the injective property. Second, when the number of MC samples is 10, we can observe an obvious margin in global alignment, which refers to the computation between mixture distributions.

Finally, we are also interested in the performance of the proposed method under challenging small data scenarios compared with baseline methods. As a result, we conduct two kinds of experiments with different conditions on skin lesion classification, including a fixed number of samples per class

Table 10: Fixed number of samples per class in each source domain.

| Number of sample per class for each source domain | DeepAll | DSU | BDIL | DNA? | Ours |
|---|---|---|---|---|---|
| 40 | 0.5399 ±0.0156 | 0.6145 ±0.0175 | 0.5897 ±0.0029 | 0.5412 ±0.0143 | **0.6368** ±0.0074 |
| 30 | 0.5309 ±0.0201 | 0.5458 ±0.0184 | 0.5762 ±0.0101 | 0.5132 ±0.0229 | **0.6138** ±0.0291 |
| 20 | 0.5044 ±0.0129 | 0.5243 ±0.0143 | 0.5573 ±0.0011 | 0.5048 ±0.0087 | **0.6037** ±0.0121 |

Table 11: Fixed fraction of samples in each source domain.

| Fraction of sample for each source domain | DeepAll | DSU | BDIL | DNA | Ours |
|---|---|---|---|---|---|
| 100% | 0.6674 ±0.0312 | 0.6935 ±0.0121 | 0.7059 ±0.0284 | 0.7121 ±0.0141 | **0.7276** ±0.0123 |
| 80% | 0.6614 ±0.0123 | 0.6717 ±0.0029 | 0.6625 ±0.092 | 0.6591 ±0.0022 | **0.6975** ±0.0036 |
| 60% | 0.6249 ±0.0122 | 0.6299 ±0.0114 | 0.6468 ±0.0106 | 0.6149 ±0.0112 | **0.6641** ±0.0114 |
| 40% | 0.5911 ±0.0215 | 0.6188 ±0.0541 | 0.6491 ±0.0171 | 0.6065 ±0.0111 | **0.6579** ±0.0057 |

in each source domain and a fixed fraction of samples in each source domain. We choose MSK dataset as the target domain and the remaining domains as the source domains.

**Fixed number of samples per class in each source domain.** Specifically, we randomly draw T samples from each class in a source domain to represent this domain for training. Here, we set T to 20,30, and 40, respectively, in different experiments. The experimental settings are the same as the descriptions in A.2, except for the training steps as 600.

The results can be found in Table 10. As we can see, our proposed method achieved the best performance among all settings compared with all baseline methods. Meanwhile, it seems that the Bayesian-based DG approaches (e.g., our proposed method and BIDL) have better performance compared with other methods, which is reasonable as the BNN can be adaptive to the small data scenario well. Especially, our proposed method has around 5% improvements compared with the second-best method when $T$ is set to smaller, i.e., 20.

**Fixed fraction of samples in each source domain.** Specifically, we randomly draw $C\%$ samples from the training samples of each source domain to represent this domain for training. For example, D7P dataset has 963 training samples originally. The total number of samples for this domain is $40\% \times 963 = 385$ for training when $C$ is set to 40%. Here, we set $C$ to 40%, 60%, and 80%, respectively, as separately different experiments. Note that $C$ can not be set too small (as the number of samples in some classes of some domains is very limited.), otherwise the batch size can not be uniform. We can observe this kind of setting is challenging as the total number of samples for each domain is gradually small. The results can be found in Table 11. We can observe from Table 11 that our proposed method also achieves a relatively stable and better performance compared with baseline methods, as the decrease of fraction of samples in each source domain.

## A.7 DISCUSSIONS

In this paper, the definition of small data is based on a specific task, including the difficulty of prediction, the quality of on-hand images, the number of source domains, and so on. Small data scenarios may be a relative concept. Specifically, a small data scenario not only can represent the number of training examples is small among all source domains compared with some large volumes of datasets but also can reflect the number of training examples is relatively smaller in some (certain) domains. On the abovementioned conditions, it may be difficult to ensure reliable contrastive semantic loss with point-wise (or local) alignment and distribution-wise (or global) alignment because both of them require sufficient samples among source domains. Our proposed method aims to improve

performance over the abovementioned small data scenarios, which is a significant motivation for this work.

