# OpenReview forum: "Domain Generalization with Small Data"
_ICLR.cc/2023/Conference — Submitted to ICLR 2023_

### Official Review · Reviewer_s3JK · 2022-10-23

**Confidence:** 5
**Correctness:** 4
**Technical Novelty And Significance:** 4
**Empirical Novelty And Significance:** 4
**Recommendation:** 8

**Clarity, Quality, Novelty And Reproducibility:**

The paper is written well. Novelty comes from looking at probabilistic extensions of classical MMD and contrastive semantic alignment loss. Experiments are clearly explained. Overall a good paper.

**Strength And Weaknesses:**

Strengths: Domain generalization for small sample cases is an important problem, especially in digital pathology. Probabilistic extensions of MMD and contrastive semantic loss make the paper more interesting. The domain of digital pathology is ideal for testing the efficacy of the proposed approach. Experimental results are convincing.
Weaknesses: The authors are not the first to come up with a measure to evaluate the discrepancy between mixture distributions (i.e., source domains). Please refer to Y. Balaji, R. Chellappa and S. Feizi, “Normalized Wasserstein for Mixture Distributions with Applications in Adversarial Learning and Domain Adaptation”, Proc. Intl. Conf. on Computer Vision, Seoul, South Korea, Oct. 2019 for a related work.

**Summary Of The Paper:**

The problem of domain generalization in the context of small samples is discussed. This is particularly important in digital pathology.
Novel probabilistic extensions of classical MMD and contrastive semantic alignment loss provide additional features. Evaluations on standard pathology datasets show reasonable performance improvements.

**Summary Of The Review:**

This paper addresses an important challenge in digital pathology, domain generalization when sample size is small. The proposed approach is solid with good experimental results and reasonable improvements over SOTA.

---

> ### Author Response · Authors · 2022-11-12
> **Response to Reviewer s3JK**
>
> **We thank the reviewer for the very positive comments of our submission. We do agree with the reviewer that our proposed method is related to mixture distribution alignment [Balaji, ICCV’19]. However, there are three main differences between [Balaji, ICCV’19] and our proposed methods**.
>
> - **Data sources**: [Balaji, ICCV’19] focused on the data which are intrinsically drawn from a mixture distribution, while the mixture distribution in our problem is associated with data sampling in Bayesian neural network, where the components of distribution are known in advance.
>
> - **Global alignment**: Since the components are known, we propose to explore the hierarchical information (i.e., distribution of distribution) during domain alignment with a novel P-MMD (implemented by using level-2 kernel) instead of introducing additional mixture proportion term which needs to be further optimized.
>
> - **Local alignment**: Besides only focusing on domain alignment (global alignment in our manuscript), we also explore the data similarity information (through contrastive learning) across different domains in a distribution alignment manner, which is also novel.
>
> We thank the reviewer again for pointing this out and have updated the discussion in our revised manuscript accordingly.

---

### Official Review · Reviewer_BRdo · 2022-10-25

**Confidence:** 4
**Correctness:** 3
**Technical Novelty And Significance:** 3
**Empirical Novelty And Significance:** 3
**Recommendation:** 5

**Clarity, Quality, Novelty And Reproducibility:**

- The paper is in general well-written and easy to understand. There are some aspects such as table captions and details around experiments that can be improved.
- The paper is more of incremental nature (i.e. Xiao et al. (2021)), however there are novel components specifically exploring a novel application in a data/application intensive manner.
- There are missing details around implementation and experimental setup. In each experiment it is not clear what are the train dataset(s) and how hyperparam selection has been done. There are pointers all around the main paper and appendix which are hard to follow, however this could significantly improve. I would also suggest adding a similar to Table 6 breakdown for all of the tasks.
- Dataset sizes are missing in some cases.
- Datasets are majorly public, so having the implementation/open source package (I cannot find any link to any source code) and more details around experiments (see above) reproducibility should be feasible.
- There is no statistically significant test analysis, although some results such as Table 5 and 1 is reporting standard deviation and ran for five times.


**Details Of Ethics Concerns:**

The paper uses the skin datasets that suffer from long-tail issues and results for such dataset and application should get analyzed per classes (to address any biases introduced for any specific class or ethnicity and race). If the access to such meta-data is not feasible authors should address concerns around fairness and bias upfront.


After Rebuttal: An ethics statement has been added as requested. Thanks.

**Strength And Weaknesses:**

**Strength:**
- The paper in general is very well-motivated and the target problem is relevant to both the ML and Medical AI community.
- They evaluate and ablate the method in multiple settings.

**Weakness:** (see below for more details)
- Paper can benefit from the redesign of experiment to better support and showcase the claim of the paper around small data.
- Some details around experimental protocol and details are either missing or not presented concretely.
- Results are marginal and I cannot observe significant improvement specifically for small datasets.

**Suggestion to improve paper:**
- One of my main concerns is around the evaluation protocol of the paper. For each application, we are dealing with multiple datasets and dataset size ranges. For example skin lesion classification task includes 7 public dataset with different dataset size as mentioned in the paper, for example UDA and PH2 includes 601 examples and 200 examples consequently. What has not been mentioned in the paper is that for example HAM1000  includes 10015 dermascopic images and more critical details around other dataset are also missing. In this experimental setting paper claims the method improves performance for small data setup where we are dealing both with the domination danger of larger dataset and also limits and boundaries of improvement is not clear.  For paper claims, I am more interested in the question of how many samples is enough to reach the baseline performance? So coming up with unification is very important, picking up the smallest dataset size, and analyzing everything in the fixed dataset size or fraction (e.g. performance with 10 samples, vs. performance with 200 samples). Such experiments are standard in representation learning literature. Does the larger dataset dominate the learned space?
- Even in the current setting I would expect to see significant improvement given the proposed method for SON, UDA and PH2 but all I am seeing is marginal improvement given standard deviation on five runs.  With this experimental protocol at best, the paper is a domain generalization paper for medical image processing, and any claim around small data should be addressed and assessed concisely.
- More importantly, as the paper mentioned some of at least skin lesion dataset suffers from class imbalance, long tail issue, and effect of any method should get analyzed per class and fairness consideration should be addressed.

**Minor Editorial:**
- Improving tables and image captions can help with the quality and readability of the tables. (e.g. bold and underline, number of runs to obtain std for table 2, do they run significant tests, if so, highlighting significant improvements, and more.)
- Figure 3 is very small, moving some results to the appendix does not hurt the paper or using alternative visualizations such as bar graphs help with the quality of results.
- Last paragraph of related work, page 3, “we noticed …” can benefit from a rewriting.


**Summary Of The Paper:**

As the title suggests, this paper explores domain generalization for small data through a probabilistic maximum mean discrepancy (MMD) approach. For this purpose a domain-invariant representation is being learned from multiple domains (sources) where the assumption is that these domains inherently have insufficient samples (a.k.a small data). For the training and evaluation purpose the paper explores their claims in the medical image processing domain (skin lesion classification, epithelium-stroma classification, spinal cord gray matter segmentation) where such data limits are more clear.  They also studied and ablated each component of the proposed method including but not limited to the effectiveness of domain-invariant loss, use of global vs. local alignment, use of Bayesian layer and such.

**Summary Of The Review:**

Although I appreciate that the paper presents various experiments, the evaluations and numbers do not support the claim of the paper around small data (no significant improvement in dataset with less data), also the evaluation and experiment protocol can improve drastically (i.e. controlled sampled size). Moreover, clarity and reproducibility of paper can be improved by reorganizing some sections and presenting experimental protocol upfront and in a concise manner.

After Rebuttal: As most of my concerns has been answered, I increased my score.

---

> ### Author Response · Authors · 2022-11-12
> **Response to Reviewer BRdo (Part 1)**
>
> **We thank you for your insightful comments, and hope our responses can address all the concerns.**
>
> **Q 1-1:** *One of my main concerns is around the evaluation protocol of the paper. For each application, we are dealing with multiple datasets and dataset size ranges. For example skin lesion classification task includes 7 public dataset with different dataset size as mentioned in the paper, for example UDA and PH2 includes 601 examples and 200 examples consequently. What has not been mentioned in the paper is that for example HAM1000 includes 10015 dermascopic images and more critical details around other dataset are also missing?*
>
> **Response:**
>
> The topic of this paper focuses on **the DG problem in the context of insufficient data**. The **reliable contrastive semantic loss** with point-wise (or local) alignment and **distribution-wise (or global) alignment** between domains **both requires sufficient samples among source domains.** However, **these sufficient samples from multiple source domains may not always be available and accessible.** For example, for the skin lesion classification **(Page 7, section 4.1),** UDA (601 examples) and SON (200 examples) domains have a small number of training images indeed, although some domains may have a relatively larger number of training images (e.g., HAM10000 with 10015 examples).
>
> **As a result, these domains which have fewer data are mentioned more, such that the experiment setting is consistent with our motivation.**
>
> **To address these missing details about datasets, we have added a table to list the number of training examples in each task for reference. More details can be found in revised manuscript (Page 18, Table 9)**

---

> ### Author Response · Authors · 2022-11-12
> **Response to Reviewer BRdo (Part 2)**
>
> **Q 1-2:**  *In this experimental setting paper claims the method improves performance for small data setup where we are dealing both with the domination danger of larger dataset and also limits and boundaries of improvement is not clear. For paper claims, I am more interested in the Q of how many samples is enough to reach the baseline performance?*
>
> **Response:**
> 1) Regarding “small data” in the problem of DG, it can be related to the total number of training data across multiple source domains and also the size of data in some specific domains. The rationality is that if we are not able to have sufficient data in some domains, **it may be difficult to ensure reliable contrastive semantic loss** with point-wise (or local) alignment and **distribution-wise (or global) alignment** (because **both of them require sufficient samples among source domains).** Our proposed method aims to improve the performance over the abovementioned small data scenario.
>
> **The choice of our experiments is consistent with our motivation**. Specifically,
>
> - For the skin lesion classification **(Page 7, section 4.1),** UDA (601 examples) and SON (200 examples) domains have a **small volume of training images** indeed, although some domain may have a **relatively larger number of training images** (e.g., HAM10000 with 10015 examples). Therefore, **this experiment setting is consistent with our motivation**. We can observe from Table 1 (**Page 7 in original paper**), **our proposed method can consistently achieve the best or the second-best results in each cross-domain task**, although some baselines can achieve the best performance in a certain task. Moreover, the average performance (**Page 7, the last column of Table 1**) of our proposed method is better than that of baseline methods **with a clear margin**.
> - For epithelium stroma classification (**Page 8**), all domains have relatively small number of  samples (**NKI: 671, VGH: 615; IHC:1376**). In this scenario, **our proposed method also achieves a better performance with a clear margin compared with baseline methods (Page 7, Table 2).**
> - **Last, we can observe from Tables 3,5,6,7 that the improvements (either in some sub-tasks or in average result) can be consistently attained.**
>
> **In the revised manuscript, we have added a discussion section about the small data setup in Appendix section to clarify this point (Page 20).**
>
> 2) For the problem about “**how many samples is enough to reach the baseline performance**?”, as discussed above, the definition of “**small data**” is based on a **specific task (task dependency for short).** More importantly, the performance can also be affected by the adopted models. As a result, it can be challenging to determine how many samples are enough to reach the baseline performance. **For our experiments, all the baseline methods adopt the same number of training examples on a fair condition.**
>
> **Here, we report the number of images of source domains on a skin lesion cross-domain task (where MSK dataset as the target domain and the remaining datasets as the source domains) as a reference**.
>
> - The accuracy of “DNA” and our proposed methods was 0.7121 and 0.7276, respectively, when we utilize 100% training data (the total number of images is around 11609) from each source domain.
> - The accuracy of “DNA” and our proposed methods was 0.6591 and 0.6975, respectively, when we utilize 80% training data (the total number of images is around 9280)  from each source domain.
> - The accuracy of “DNA” and our proposed methods was 0.6149 and 0.6641, respectively, when we utilize 60% training data (the total number of images is around 6900) from each source domain.
> - The accuracy of “DNA” and our proposed methods was 0.6065 and 0.6579, respectively, when we utilize 40% training data (the total number of images is around 4600) from each source domain.
>
> As we can see,  the performances of the baseline method and our proposed method both have a drop when the number of samples decreases. Nevertheless, our proposed method can achieve better performance compared with baseline methods in all different settings with different number of training samples, which shows the effectiveness of our proposed method.

---

> ### Author Response · Authors · 2022-11-12
> **Response to Reviewer BRdo (Part 3)**
>
> **Q 1-3:** *So coming up with unification is very important, picking up the smallest dataset size, and analyzing everything in the fixed dataset size or fraction (e.g. performance with 10 samples, vs. performance with 200 samples). Such experiments are standard in representation learning literature. Does the larger dataset dominate the learned space?*
>
> **Response:**
>
> 1) By following your suggestions, we conducted new experiments on skin lesion classification task, where we choose MSK as  the target domain and the remaining domains as the source domains.
>
> - **Fixed number of samples per class in each source domain.** Specifically, we randomly draw *T* samples from each class in a source domain to represent this domain for training. Here, we set *T* to 20,30, and 40, respectively, in different experiments. The results can be found as following:
>
>
>
>
> Table 1 **Fixed number  of samples per class in each source domain**
> | **Number of sample  per class for each source domain** | **DeepAll** | **DSU** | **BDIL** | **DNA** | **Ours** |
> |:---:|---|---|---|---|---|
> | **40** | 0.5399$\pm$0.0156 | 0.6145$\pm$0.0175 | 0.5897$\pm$0.0029 | 0.5412$\pm$0.0143 | **0.6368$\pm$0.0074** |
> | **30** | 0.5309$\pm$0.0201 | 0.5458$\pm$0.0184 | 0.5762$\pm$0.0101 | 0.5132$\pm$0.0229 | **0.6138$\pm$0.0291** |
> | **20** | 0.5044$\pm$0.0129 | 0.5243$\pm$0.0143 | 0.5573$\pm$0.0011 | 0.5048$\pm$0.0087 | **0.6037$\pm$0.0121** |
>
> As we can see, our proposed method achieved the best performance among all settings compared with all baseline methods. Meanwhile,  it seems that the Bayesian-based DG approaches (e.g., our proposed method and BIDL) have better performance compared with other methods, which **is reasonable as the BNN can be adaptive to the small data scenario well**. Our proposed method has ~5% improvement compared with the second-best method when T is set to smaller, i.e., 20.
>
> - **Fixed fraction of samples in each source domain.** Specifically,** we randomly draw *C%* samples from the training samples of each source domain to represent this domain for training. For example, D7P dataset has 963 training samples originally. The total number of samples for this domain is 40%\**963=385* for training when C is set to 40% . Here, we set *C* to 40%, 60%, and 80%, respectively, in different experiments. Note that *C* can not be set too small (as the number of sample in some classes of some domains is very limited.), otherwise the batch-size can not be uniformed. We can observe this kind of setting is challenging as the  total number of samples for each domain is gradually small. The results can be found as following:
>
> Table 2 **Fixed fraction of samples in each source domain.**
> | **Fraction of sample   for each source domain** | **DeepAll** | **DSU** | **BDIL** | **DNA** | **Ours** |
> |:---:|---|---|---|---|---|
> | **100%** | 0.6674 $\pm$0.0312 | 0.6935 $\pm$0.0121 | 0.7059 $\pm$ 0.0284 | 0.7121 $\pm$ 0.0141 | **0.7276 $\pm$ 0.0123** |
> | **80%** | 0.6614 $\pm$ 0.0123 | 0.6717 $\pm$ 0.0029 | 0.6625 $\pm$0.092 | 0.6591 $\pm$ 0.0022 | **0.6975 $\pm$ 0.0036** |
> | **60%** | 0.6249 $\pm$ 0.0122 | 0.6299 $\pm$ 0.0114 | 0.6468 $\pm$ 0.0106 | 0.6149 $\pm$ 0.0112 | **0.6641 $\pm$ 0.0114** |
> | **40%** | 0.5911 $\pm$ 0.0215 | 0.6188 $\pm$ 0.0541 | 0.6491 $\pm$ 0.0171 | 0.6065 $\pm$ 0.0111 | **0.6579 $\pm$0.0057** |
>
>
> As we can see, our proposed method also achieved the best performance among all settings compared with all baseline methods, which further shows the effectiveness of our proposed method.
>
> 2) For the concern about “**Does the larger dataset dominate the learned space**?”, some points need to be clarified as following:
> **The objective of DG is to learn a domain-invariant representation, where the learned feature representation is expected to be shareable across different domains. As such, the overfitting problem where “large dataset dominate the learned space” can be mitigated as we require the learned model to be better generalized to unseen but related domain during testing**.

---

> ### Author Response · Authors · 2022-11-12
> **Response to Reviewer BRdo (Part 4)**
>
> **Q 2-1:** *Even in the current setting I would expect to see significant improvement given the proposed method for SON, UDA and PH2 but all I am seeing is marginal improvement given standard deviation on five runs.*
>
> **Response:**
>
> - Our proposed method can consistently achieve the best or the second-best results in each cross-domain task, although some baselines can achieve the best performance in a certain task. For example, our proposed method can significantly achieve better performance than DNA in D7P task (**our proposed method: 0.6120 *v.s.* DNA: 0.5581**), although the improvement achieved  by our proposed method may be marginal (as mentioned by you) than DNA for UDA task (**our proposed method: 0.7486 *v.s.* DNA: 0.7314**). More importantly, the average performance (**Page 7, the last column of Table 1**) of our proposed method is better than that of baseline methods with a clear margin.
>
> For the epithelium stroma classification task from Table 2 (**Page 7**), our proposed method also achieve ~3% improvement compared with the second-best baseline methods. Our proposed method achieve the best results among all tasks with a clear margin.
>
> - From the additional experiments in Table 1 (**Fixed number of samples in per-class in each source domain**) and Table 2 (**Fixed fraction of samples in each source domain**), the obvious improvements achieved by our proposed method can be observed  compared with baseline methods on very challenging small-data DG tasks.
> - Last but not the least, the proposed method in this paper is a general framework that can be implemented in different tasks (medical imaging datasets and benchmark datasets for natural images). Extensive experiments show the effectiveness of our proposed method.
>
> **Q 2-2:** *With this experimental protocol at best, the paper is a domain generalization paper for medical image processing, and any claim around small data should be addressed and assessed concisely.*
>
> **Response:**
>
> - We kindly disagree that our proposed method is “**domain generalization paper for medical image processing at best**”. **Actually, the proposed method is a general framework, which can be utilized in  different tasks. Extensive experiments on medical imaging datasets and widely-adopted benchmark datasets (natural  images, PACS and OfficeHome, see Table 5 in Page 9 and Table 7 in Page 14) show the effectiveness of proposed method. Besides, two novel P-MMD (a probabilistic extension of classical MMD) and P-CSA measures are proposed. They may be used to measure the distance between distribution-related data well.  Meanwhile, our proposed method focuses on the DG problem in the context of  small data.**
>
>
> - **Regarding the claim of “small data” for the problem of DG, we have discussed the motivation in the introduction section that** if we are not able to have sufficient data in some domains, **it may be difficult to ensure reliable contrastive semantic loss** with point-wise (or local) alignment and **distribution-wise (or global) alignment** (because **both of them require sufficient samples among source domains).**
>
> **Based on this motivation, we can have two different scenarios associated with DG.**
>
> - **On the one hand, “small data” scenario can represent the number of training examples is small  among all domains compared with some big volume of (or “larger”, as mentioned by you) datasets.** For example, for the skin lesion classification **(Page 7, section 4.1),** UDA (601 examples) and SON (200 examples) domains have a small number of training images indeed, although some domain may have a relatively larger number of training images (e.g., HAM10000 with 10015 examples). Therefore this experiment setting is consistent with our motivation. Our proposed method shows better performance compared with existing baseline methods.
>
> - **On the other hand, “small data” scenario also can represent the number of training examples is small  among all domains. It is also difficult to provide a reliable contrastive semantic alignment and distribution-wise alignment.** For example, Epithelium Stroma classification (**Page 8**), all domains have a relatively small number of  samples (**NKI: 671, VGH: 615; IHC:1376**). In this scenario, **our proposed method also achieves a better performance with a clear margin compared with baseline methods.**
>
> - **Besides these two scenarios, we also design new experiments on the fixed small-data conditions  (in the Response to Question 1-3), where the results further justify the effectiveness of our proposed method.**

---

> ### Author Response · Authors · 2022-11-12
> **Response to Reviewer BRdo (Part 5)**
>
> **Q 3:** *More importantly, as the paper mentioned some of at least skin lesion dataset suffers from class imbalance, long tail issue, and effect of any method should get analyzed per class and fairness consideration should be addressed.*
>
> **Response:** First of all, we want to argue that class imbalance and data bias are still open-problems in the community of machine learning. **In this paper, we focus more on the DG problem in the context of small data. Specifically, we hope a model can generalize better to the unseen but related data during testing under insufficient source domain data, which is an urgent topic. While we do agree with reviewer that the problem of long tail issue and class imbalance are important, they are beyond the scope of our submission since we focus on designing a general DG framework. In this work,**  we follow existing training protocol [Li, NeurIPS’2020] on skin lesion to validate the effectiveness of the proposed method.  The superiority of our proposed method was validated well though the accuracy.
>
> **Besides skin lesion classification, we also consider additional experiments,** including epithelium stroma classification, spinal cord GM segmentation, and two popular benchmark datasets for natural images consistently reflect the general effectiveness of  the proposed method.
>
> **Q 4:** *Improving tables and image captions can help with the quality and readability of the tables. (e.g. bold and underline, number of runs to obtain std for table 2, do they run significant tests, if so, highlighting significant improvements, and more.)*
>
> **Response:** Thanks for your suggestions. In the revised manuscript, we have followed your suggestions to improve the readability.
>
>
>
> **Q 5:** *Last paragraph of related work, page 3, “we noticed …” can benefit from a rewriting*.
>
> **Response:** Thanks for your suggestions. In the revised manuscript, we have rewritten this paragraph.
>
> **Q 6:** *The paper is more of incremental nature (i.e. Xiao et al. (2021)), however there are novel components specifically exploring a novel application in a data/application intensive manner.*
>
> Response: In *Xiao et al. (2021)*, the authors proposed to consider the uncertainty of a generalizable model based on BNN, where the distances of positive probabilistic embedding pairs and class distribution are minimized via KL measure. Despite the effectiveness, the dissimilar pairs (i.e., negative pairs) are ignored, which may not benefit feature representation learning. Moreover, they only focused on sample similarity while the distribution information is ignored. Instead, our proposed method comprehensively considers both positive and negative probabilistic embedding pairs via a novel distribution-based contrastive semantic loss. More importantly, our proposed method can marriage the measurement on the distribution over distributions (i.e., the global perspective alignment) and the distribution-based contrastive semantic alignment (i.e., the local perspective alignment).
>
> **Q 7:** *There are missing details around implementation and experimental setup. In each experiment it is not clear what are the train dataset(s) and how hyperparam selection has been done.*
>
> **Response:**
>
> - **For common DG settings, the referred domain of each column in each table usually denotes that this domain is the target domain and the remaining are source domains (as training datasets).** For example, for Table 1 in Section 4.1, the second column denotes that the “DMF“ dataset is the target domain and the source domains are remaining ones (including D7P, MSK, PH2, SON, UDA and HAM10000). Similar rules are followed by Table 2, 4,5,7. We are sorry that the descriptions may not be clear for the reader. **In the revised manuscript,  we have added corresponding descriptions in each caption of the Table, in order to make the reader understand the information of training datasets clearly.**
>
> - For the hyperparameter selection, we turn the hyperparameters on spited validation set for each task. In the revised manuscript, the additional description is added.
>
>
>
> **Q 8:** *There are pointers all around the main paper and appendix which are hard to follow, however this could significantly improve. I would also suggest adding a similar to Table 6 breakdown for all of the tasks.*
>
> **Response:** Thanks for your suggestion. In the revised manuscript, we have followed your advice to make the table clearer.
>
> **Q 9:** *Dataset sizes are missing in some cases.*
>
> **Response:** To address this problem in the revised manuscript, we have added a table to list the number of training examples in each task for reference (**Page 18, Table 9**)**.**
>
>
> **Q 10:** *There is no statistically significant test analysis, although some results such as Table 5 and 1 is reporting standard deviation and ran for five times.*
>
> **Response:** we follow previous works reporting std, which is also common practice in the community of machine learning.

---

> ### Author Response · Authors · 2022-11-12
> **Response to Reviewer BRdo (Part 6)**
>
> **Q 11:** *The paper uses the skin datasets that suffer from long-tail issues and results for such dataset and application should get analyzed per classes (to address any biases introduced for any specific class or ethnicity and race). If the access to such meta-data is not feasible authors should address concerns around fairness and bias upfront.*
>
> **Response:**
>
> - **For the first concern:**
>
> We agree with the reviewer that the long-tail and data bias issues of skin datasets are important. In our submission, we focus on the problem of domain generalization instead of long-tail/imbalance data classification. We therefore follow the previous work [LI, NeurIPS’20] on skin datasets, and the superiority of the proposed method was validated well. Besides, other cross-domain tasks, including epithelium stroma classification, spinal cord GM segmentation, and two popular benchmark datasets for natural images consistently reflect the effectiveness of  the proposed method.
>
> - **For the second concern:**
>
> The access of skin datasets as well as other datasets (including epithelium-stroma dataset and spinal cord GM dataset) is feasible if you sign a corresponding agreement with the provider or download open-source datasets on the given website directly. **No personal or private information exists in these obtained datasets. Here, the links of used medical imaging dataset  are provided as**
>
> Spinal Cord GM segmentation: <http://niftyweb.cs.ucl.ac.uk/challenge/index.php>
>
> Epithelium-stroma dataset: <http://fimm.webmicroscope.net/supplements/epistroma>
>
> Skin lesion  dataset: <https://challenge.isic-archive.com/landing/2018/47/>
>
> **In the revised manuscript, we have added a Ethics Statement (in Page 10) section (following the policy of ICLR conference) to clarify potential concerns about ethics.**

---

> ### Comment · Reviewer_BRdo · 2022-11-17
> **Acknowledge and Thanks**
>
> Just to acknowledge that I have read your response and have no further questions.
> Thanks for the effort to address my concerns.

---

> > ### Author Response · Authors · 2022-11-21
> > **Thanks**
> >
> > Dear Reviewer BRdo,
> >
> > Thanks again for your insightful comments in the first round. We noticed that you have raised your score from 3 to 5, as you have no further questions and most of your concerns have been addressed. May we know the specific reason you still rate our submission as “below the acceptance bar“ ?
> >
> > We will do our best to clarify your concerns.
> >
> > Best,
> >
> > The authors

---

### Official Review · Reviewer_6Mag · 2022-10-25

**Confidence:** 3
**Correctness:** 2
**Technical Novelty And Significance:** 3
**Empirical Novelty And Significance:** 2
**Recommendation:** 5

**Clarity, Quality, Novelty And Reproducibility:**

[Clarity]
The current manuscript is well-written and easy to follow, except for the experimental conditions.

[Quality]
Although the proposed method has a potential, I am not sure whether the proposed method is a better solution for the targeted problem.

[Reproducibility]
I should say that the current manuscript does not have sufficient reproducibility due to the lack of detailed experimental conditions.

**Strength And Weaknesses:**

[Strength]

S1. The problem dealt with in this paper is significant. Domain generalization is one of the fundamental problems in machine learning and computer vision, and in some cases, we cannot obtain sufficient training examples for training machine learning models for this purpose. Technically solid methods with thorough experimental evaluations will draw attention from a broad range of researchers and engineers.

S2. The use of level-2 kernels for extending the standard MMD is interesting. The proposed p-MMD might be useful for other tasks emphasizing probabilistic distributions.

[Weakness]
W1. The proposed method lacks "why". I understand that the proposed method extends the standard MMD by introducing a kernel function for kernel means and it seems to be novel as far as I know. However, I am unsure why the proposed method can boost the performance on few-resource domain generalization.

W2. The current main content lacks sufficient experimental conditions. For example,
- What is the source / target domain data in the experiment presented in Section 4.1? --> This makes the task setting totally unclear.
- How many training examples are available in the experiment presented in Section 4.1? --> I could not understand whether the current experimental setting can be regarded as "few resources" or not.
- How many competitors include the Bayesian framework in them? --> I could not understand which contributes to the performance improvement, Bayesian neural networks or probabilistic MMD.

**Summary Of The Paper:**

This paper deals with the problem of domain generalization in the case that we have only insufficient source domain examples and proposes a new loss function called probabilistic maximum mean discrepancy (p-MMD). The proposed loss function introduces a kernel function \kappa for kernel means of latent embeddings and applies a standard MMD for the kernel \kappa. If the kernel \kappa and a kernel k for latent embeddings z are both Gaussian, the proposed loss function can be computed by the kernel k.

**Summary Of The Review:**

I am not so positive about this paper due to the weaknesses presented in this review, namely (1) the mismatch between the targeted problem and the proposed method and (2) the lack of significant experimental conditions. I believe that the proposed probabilistic MMD would be useful for some tasks. I strongly recommend the authors reconsider problems for which the proposed method is suitable.

After author feedback: I acknowledge the feedback; however, I would keep the original review score.
- I agree with Reviewer BRdo in terms of the redesign of experiments.
- Significant experimental conditions should be clearly presented in the main content. Or, some pointers to the appendices should be presented in the main content.

---

> ### Author Response · Authors · 2022-11-12
> **Response to 6Mag (Part 1)**
>
> **We thank you for your positive comments, and hope our responses can address your concerns.**
>
> **Q1:** *The proposed method lacks "why". I understand that the proposed method extends the standard MMD by introducing a kernel function for kernel means and it seems to be novel as far as I know. However, I am unsure why the proposed method can boost the performance on few-resource domain generalization.*
>
> **Response:** Thanks for reviewer’s comments. The reasons why the proposed method can improve the performance of domain generalization on small data come from three aspects:
>
> - **BNN can be adaptive to small data well compared with deterministic models (e.g., CNNs).**
>
> In the context of small data, BNN usually has better generalization performance, as the probabilistic model can learn a distribution over model weights that can incorporate the uncertainty into the overall model when there is insufficient data. Quite a few works reported better performance compared with deterministic models, such as predicting the data directly [Graves, NeurIPS’11] and improving the quality of captured feature embeddings from high-dimensional insufficient inputs [Ankur, UAI’ 21].
>
> For our proposed method, the BNN is introduced to the DG problem in the context of insufficient data, where BNN-based feature extraction and classification layers can take both consistent improvements through our ablation experiments (**In original paper,** **Page 8, the second and third rows in Table 4; Page 9, the last second row in Table 5**).
>
> **Moreover, to explore a better performance for DG problem with small data, it is much imperative to design a new and adaptive methodology for the probabilistic framework such that domain-invariant representation (this is a common strategy for DG problem) can be learned well.** The domain-invariant learning in the sequel can benefit from the improvement of the quality of captured probabilistic embeddings from insufficient inputs.
>
> **Reference**
>
> Alex, Graves. "Practical variational inference for neural networks." NeurIPS (2011).
>
> Mallick, Ankur, et al. "Deep kernels with probabilistic embeddings for small-data learning." UAI, 2021.
>
> - **Domain-invariant representation learning under a probabilistic framework from global and local perspectives.**
>
> Many DG researches (see this review [Zhou, TPAMI’22]) have shown that domain-invariant representation learning for embeddings can improve the generalization performance for DG problem. However, how to perform domain-invariant feature representation learning is underexplored. For our probabilistic embeddings, an unavoidable difficulty is how to learn domain-invariant representation from a global perspective (i.e.,  minimizing the discrepancy among domains). **Classical MMD estimate may not be directly utilized to measure the discrepancy between two source domains for our probabilistic embeddings,** as a source domain in latent space consists of a serial of latent distributions (**At a high level, a source domain can be regarded as a distribution over distributions**) rather than latent points.
>  As a result, we extend the **classical MMD** estimate to a **probabilistic version** (**P-MMD**) via imposing a *level-2* kernel, which can measure two distributions over distributions more accurately, as the distributions of means, variances and higher moments of the sub-distributions associated to points of different source domains can be considered (**Details are discussed in Page 5, Corollary 2 and its Remarks, in original paper**).  Note that we also compare **our proposed P-MMD** with the method that indirectly uses the **classical** MMD through pre-processing the probabilistic embeddings into points with simple mean operation. Better performance can be achieved by the proposed P-MMD (**Appendix section, Page 18, A6**) compared with simple mean operation (More hierarchical information is sacrificed). The ablation experiments (**Page 8, the last second row in Table 4**) also show that the cross-domain performance can be improved by equipping the global alignment (endowed by P-MMD), compared with BNN-based baseline (**Page 8, the third row in Table 4**).
> Similar improvements for domain-invariant learning from local perspective can also be observed (**Page 8, the last third row in Table 4**) through imposing **proposed probabilistic contrastive semantic alignment(P-CSA).**
>
> **Reference**
>
> Zhou, Kaiyang, et al. "Domain generalization: A survey." TPAMI (2022).
>
> - **Last but not the least, the reasons why the proposed method can improve the performance on domain generalization on small data benefit from the integration between the probabilistic framework and corresponding probabilistic-endowed domain-invariant learning (where two novel P-MMD and P-CSA components are proposed). The ablation experiments (on Page 8, the last row in Table 4) show that the cross-domain performance can be improved by this point.**

---

> ### Author Response · Authors · 2022-11-12
> **Response to Reviewer 6Mag (Part 2)**
>
> **Q2:** *The current main content lacks sufficient experimental conditions.*
>
> **Response: Due to the page limit,** most experimental settings for each task were reported in **Appendix section (From Page 15 to Page 18).** The datasets and implement details are also presented specifically there, including the **model structure, hyperparameters, and training protocols and so on**.
>
> **Q2-1:** *What is the source / target domain data in the experiment presented in Section 4.1? --> This makes the task setting totally unclear.*
>
> **Response: For Section 4.1, we followed [LI, NeurIPS’20] by reporting the results where the domain of each column in the table denotes that this domain is the target domain and the remaining domains are source domains.**
>
> For example, for Table 1 in Section 4.1, the second column denotes that the “DMF“ dataset is the target domain and the source domains are remaining ones (including D7P, MSK, PH2, SON, UDA and HAM10000). Similar rules are followed by Table 2, 4,5,7. Specially, the detailed results of each task (including source and target domains) for spinal cord GM segmentation can be clearly observed in Table 6.
>
> **We have added the information on the caption of each table in the revised manuscript such that the reader can understand the source and target domain easily.**
>
> **Reference**
>
> Haoliang Li, et al. Domain generalization for medical imaging classification with linear-dependency regularization. NeurIPS, 33:3118–3129, 2020.
>
> **Q2-2:** *How many training examples are available in the experiment presented in Section 4.1? --> I could not understand whether the current experimental setting can be regarded as "few resources" or not.*
>
> **Response:**
> 1) Due to the page limit, we cited corresponding reference (that can give the details of the number of training examples) and hope the reader can attain the dataset details from references.
>
> Here, we clarify this point as following.
>
> - **The number of training examples for skin lesion classification follows the training protocol in [Li et al. 2020], where**  HAM10000: 5007 images; DMF: 606 images; D7P:963 images; UDA: 308 images; MSK: 1959 images; PH2: 100 images; SON: 4625 images.  Note that the training examples contain  50% of total images in each domain. The remaining 20%  and 30% images in each domain are testing and validation sets, respectively.
> - **The number of training examples for epithelium and stroma classification**: IHC: 1074 image patches; NKI: 984 image patches; VGH: 1100 image patches; Note that the training examples contain  80% of total image in each domain. The remaining 20% image patches in each domain are validation set. The testing on target domain is the overall images in each domain.
> - **The number of training examples for spinal cord GM segmentation follows the training protocol in [Li et al. 2020], where** “Site 1”: 27 slices; “Site 2”: 100  slices;“Site 3”: 246 slices;“Site 1”: 122 slices;
>
> **To address this problem in the revised manuscript, we have added a table to list the number of training examples in each task for reference in page 18 (Appendix, Table 9)**
>
> 2) Regarding your comment **whether the current experimental setting can be regarded as "few resources" or not (This concern is also related to your concern “the mismatch between the targeted problem and the proposed method”)**,  we argue that our experimental settings are clear to reflect the “small data” (a.k.a. “few resources” mentioned by you) scenario from two aspects:
>
> - **The** **contrastive semantic loss** with point-wise (or local) alignment and **distribution-wise (or global) alignment** between domains **both requires sufficient samples among source domains.** However, **these sufficient samples from *multiple source domains* may not always be available or accessible (as discussion in Introduction section, Page 2)**. For example, for the skin lesion classification **(Page 7, section 4.1),** UDA (601 examples) and SON (200 examples) domains have a small number of training images indeed, although some domains have a relatively larger number of training images (e.g., HAM10000 with 10015 examples). Therefore this experiment setting is consistent with our motivation. Our proposed method shows better performance compared with existing baseline methods.
> - **“Small data” scenario also can represent the case that the number of training examples is small  among all domains. It is also difficult to provide a reliable contrastive semantic alignment and distribution-wise alignment.** For example, Epithelium Stroma classification (**on Page 8**), all domains have a relatively small number of samples (NKI: 671, VGH: 615; IHC:1376). In this scenario, **our proposed method also achieves a better performance with a clear margin compared with baseline methods.**
>
> **To sum up, the abovementioned analysis shows that the target problem and the proposed method are matched well through comprehensive small data settings.**

---

> ### Author Response · Authors · 2022-11-12
> **Response to Reviewer 6Mag (Part 3)**
>
> **Q2-3:** *How many competitors include the Bayesian framework in them? --> I could not understand which contributes to the performance improvement, Bayesian neural networks or probabilistic MMD*.
>
> **Response**:
>
> 1) For the competitors, **BDIL** is the Bayesian-based DG method.
>
> - As we discussed in the **Page 7, Second Paragraph, Line 12 in original paper,** i.e.,** “ *Benefiting from probabilistic framework as an implicit regularization, **our proposed method and BDIL** can learn a distribution over weights, which can handle insufficient samples flexibly. However, it can be observed that additional invariant classifier learning on **BDIL** and MASF may cause negative effects (see the results on DMF) on very challenging data, which may be reasonable as the explicit alignment on the classifier with high error probability can lead to negative transfer (take more uncertainty).”*,   an obvious improvement in terms of average performance can be achieved by our proposed method (**0.7328**) compared wth BDIL (**0.7049**), in skin classification task.
> - As we discussed in the **Page 8, Second Paragraph, Line 8 in original paper,** i.e., **“***Second, **BDIL** not only adopts two-level alignments on feature extractor and classifier, but also obtains further improvements by probabilistic framework. However, one can observe that **BDIL** has similar performance drop (in the skin lesion classification) on the challenging task (i.e., the NKI task).”,* an obvious improvement in terms of average performance also can be achieved by our proposed method (**0.8405**) compared wth BDIL (**0.8105**), in Epithelium Stroma Classification.
> - It would be better to highlight again our advantage. Our proposed method benefits from the probabilistic framework endowed by BNN for insufficient data from source domains. More importantly,  we propose to extend empirical MMD to a novel probabilistic MMD (P-MMD) that can empirically measure the discrepancy between mixture distributions. A novel probabilistic contrastive semantic alignment (P-CSA) loss with kernel mean embedding is proposed to encourage positive probabilistic embedding pairs closer while pulling other negative ones apart.
>
> 2) **For the contribution between the BNN and P-MMD**, **ablation study was presented in the original paper**. In **Page 8, Table 4,** we can find that the BNN **(second and third rows in Table 4**) and P-MMD (**the last second row in Table 4 for global alignment**) both can take consistent improvements. By comparing the BNN and P-MMD, the introduction of P-MMD for global alignment for plain BNN-based model further boosts the performance (**especially for the CC and DSC metrics**).
>
> **In the revised paper, we have discussed improvements in more detail in ablation study section (Page 9, 4.4).**
>
> **Q 3:**  *I should say that the current manuscript does not have sufficient reproducibility due to the lack of detailed experimental conditions.*
>
> **Response:** For the experimental settings, most experimental conditions are reported in detail in **Appendix section**, due to the **page limit for ICLR conference**. The experimental conditions can be found as follow:
> - **The Skin Lesion Classification task:** The dataset and implement detail can be found in **Appendix section, A.2,** entitled as **“EXPERIMENTAL DETAILS OF SKIN LESION CLASSIFICATION” (Page 15 in original paper).**
> - **The Epithelium Stroma Classification task:** The dataset and implement detail can be found in **Appendix section, A.3,** entitled as **“EXPERIMENTAL DETAILS OF EPITHELIUM STROMA CLASSIFICATION ” (Page 16 in original paper).**
> - **The Spinal Cord GM Segmentation task:** The dataset and implement detail can be found in **Appendix section, A.4,** entitled as **“EXPERIMENTAL DETAILS OF SPINAL CORD GRAY MATTER SEGMENTATION ” (Page 16 in original paper).**
> - **PACS and OfficeHome Benchmark experiments.**  The dataset and implement detail can be found in **Appendix section, A.5,** entitled as **“EXPERIMENTAL SETTINGS AND ADDITIONAL RESULTS ON BENCHMARKS” (Page 17 in original paper).**
>
> **We will also release our code if our paper can be accepted.**

---

> ### Author Response · Authors · 2022-12-12
> **Response to Reviewer 6Mag' Acknowledgements about the redesigned experiments and experimental conditions**
>
> Dear Reviewer 6Mag,
>
> Thank you for your acknowledgment to our response. For your two concerns, some points need to be clarified.
>
> - First, the redesigned experiments mentioned by Reviewer BRdo had been conducted and presented in the revised manuscript (Please see Appendix section-A.6 for more details). Moreover, Reviewer BRdo acknowledged that“most of my concerns has been answered and have no further questions, I increased my score.”
>
> - Second, we will provide some pointers (e.g., hyperlink) to the appendices in the main content in the future revised manuscript (as we are not able to revise the manuscript at this moment). Note that we found that the time when your acknowledgment posted was 27 Nov 2022. Any revision of the paper is not allowed in that time.
>
> Thanks a lot,
>
> The authors

---

### Official Review · Reviewer_RmYa · 2022-10-25

**Confidence:** 2
**Correctness:** 3
**Technical Novelty And Significance:** 3
**Empirical Novelty And Significance:** 3
**Recommendation:** 6

**Clarity, Quality, Novelty And Reproducibility:**

The writing overall seems clear but does require more proofreading. I'm not very familiar with other works in this field so cannot judge on the novelty of the approach.

**Strength And Weaknesses:**

The paper proposes some new methodology and shows promising results on medical datasets. But it is missing a discussion on potential drawbacks of this approach such as e.g. slower inference time. It is also missing comparison against baselines on datasets such as VLCS, TerraIncognita and DomainNet. While some of these might not showcase the strength of the proposed approach, but the expectation should be that the approach performs similarly on these compared to existing baselines.

**Summary Of The Paper:**

The paper proposes to learn a domain-invariant representation based on the probabilistic framework by mapping each data point into probabilistic embeddings. For this, a probabilistic MMD and a probabilistic CSA loss are proposed.

**Summary Of The Review:**

Overall the paper seems to propose a novel approach with promising results. The quality of the paper can be improved by addressing potential shortcomings and an even more thorough evaluation on other benchmark datasets.

---

> ### Author Response · Authors · 2022-11-12
> **Response to Reviwer RmYa**
>
> **We thank your insightful comments, and hope our responses clarify all the concerns.**
>
> **Q1:** *The paper proposes some new methodology and shows promising results on medical datasets. But it is missing a discussion on potential drawbacks of this approach such as e.g. slower inference time.*
>
> **Response:** We do agree that there exists limitations in our proposed method. For example, our framework requires longer time during training and inference due to  some additional operations/ computations in our proposed framework and MC sampling. We have addressed this issue from two aspects:
>
> - As discussed in the original submitted manuscript (**Page 5, last Line 2 to Page 6, Line 6**), we proposed to relax the time complexity *O(n^2)* for P-MMD computation to *O(n)* by drawing pairs from two domains with replacement, which is supported by **the linear statistic theory of MMD [Muandet, 2017]**.
> - As discussed in the original submitted manuscript (**Appendix, Page 18, A6, the first point**), we considered **balancing the number of Monte Carlos (MC) samples and the computational efficiency**. The experiments by using different MC samples, were conducted to explore a balanced MC sample. A promising performance and lower MC samples can be achieved when the number of MC samples is set to 10.
>
> **By following your suggestion**,
>
> - We add some discussions about the number of MC samples with computational in revised paper (**Page 4, section 3.2-last Line 2 to last Line 1**).
>
> **Reference**
>
> Muandet, Krikamol, et al. "Kernel mean embedding of distributions: A review and beyond." *Foundations and Trends® in Machine Learning* 10.1-2 (2017): 1-141.
>
> **Q2:** *It is also missing comparison against baselines on datasets such as VLCS, TerraIncognita and DomainNet. While some of these might not showcase the strength of the proposed approach, but the expectation should be that the approach performs similarly on these compared to existing baselines.*
>
> **Response:** We agree that the validation on DG benchmark datasets (such as mentioned VLCS, TerraIncognita and DomainNet) is necessary. Actually, two popular benchmark datasets, including **PACS (Page 9, section 4.4-Results on DG benchmark)**  and **OfficeHome (Appendix section, Page 17, A.5)**, **had been utilized to show the effectiveness of the proposed method in the original paper**.
>
> As we can see from Table 5 (**Page 9**) and Table 7 (**Page 17**), better performance can be achieved by our proposed method compared with state-of-the-art baselines (e.g., DNA and SWAD).
>
> **By following your suggestion**,
>
> - **We further validate the effectiveness of our proposed method on VLCS benchmark dataset as mentioned by you**. The performance against existing baselines can be found as following. We can observe that a competitive performance can be achieved by our proposed method. This result is similar with what we achieved on PACS and OfficeHome benchmarks.
>
> | Algorithm | Caltech101 |  LabelMe |   SUN09  |  VOC2007 |    Avg   |
> |:---------:|:----------:|:--------:|:--------:|:--------:|:--------:|
> | Mixstyle  | 98.3       | 64.8     | 72.1     | 74.3     |   77.4   |
> | RSC       | 97.9       | 62.5     | 72.3     | 75.6     |   77.1   |
> | DANN      | **99.0**   | 65.1     | 73.1     | 77.2     |   78.6   |
> | GroupDRO  | 97.3       | 63.4     | 69.5     | 76.7     |   76.7   |
> | MTL       | 97.8       | 64.3     | 71.5     | 75.3     |   77.2   |
> | VREx      | 98.4       | 64.4     | 74.1     | 76.2     |   78.3   |
> | MLDG      | 97.4       | 65.2     | 71.0     | 75.3     |   77.2   |
> | SagNet    | 97.9       | 64.5     | 71.4     | 77.5     |   77.8   |
> | CORAL     | 98.3       | **66.1** | 73.4     | 77.5     |   78.8   |
> | SWAD      | 98.8       | 63.3     | 75.3     | 79.2     |   79.1   |
> | DNA       | 98.8       | 63.6     | 74.1     | 79.5     |   79.0   |
> | **Ours**      | 98.9       | 63.4     | **75.8** | **79.8** | **79.5** |
>
>
> - The experimental results o VLCS benchmark dataset are presented (**Page 16**)
>
> ` `and discussed (**Page 17**) in Appendix section of the revised paper.

---

### Author Response · Authors · 2022-11-15
**General Response: Regarding the revision**

Dear reviewers,

We have updated our paper with a revision. In this version, we improve our manuscript in the following aspects:
- **Additional discussions about the targeted problem and the rationality of our proposed method are provided**, as suggested by reviewers 6Mag and BRdo.
- **Additional references are provided, and some related works have been reorganized**, as suggested by reviewers BRdo and s3JK.
- **More details about the experimental conditions/settings are added**, as suggested by  reviewers 6Mag and BRdo.
- **Additional experimental results on VLCS dataset  and Skin Lesion Classification (showcases the effectiveness of the proposed method on more challenging and unified small data scenarios) are provided**, as suggested by reviewers BRdo and RmYa.
- **The readability of tables is improved**, as suggested by reviewer BRdo.

If you have any further comments, please let us know.

Many thanks,

Authors.

---

### Decision · Program_Chairs · 2023-01-20

**Decision:**

Reject

**Justification For Why Not Higher Score:**

The 3 active reviewers - and who provided the most informative reviews and feedbacks - came to a consensus that the paper still needs an important revision. Reject was correct for all of them.
I discard review s3JK, the review was too high level and the reviewer did not participate to discussion. This is in agreement with SAC.

**Justification For Why Not Lower Score:**

N/A

**Metareview: Summary, Strengths And Weaknesses:**

This paper studies the problem of domain generalization in the presence of small data. It takes the assumption that source domaines have   insufficient amount of examples. The paper uses a new divergence which is as an extension of maximum mean discrepancy (MMD) called probabilistic MMD (p-MMD) which is applied on mixtures of distributions. This loss is used for defining a new probabilistic contrastive semantic alignment loss which allows to work on probabilistic embeddings to deal with the lack of data.

Strengths:
-The problem is important
-The proposed MMD extension is interesting

Weaknesses:
-The paper lacks of clarity and justifications in some aspects
-the experimental evaluation has many weaknesses: setup, lack of details and baselines notably.

During rebuttal, authors have provided multiple answers and additional information.
In the evaluation, I discarded review s3JK which was assessed as being over-estimated without precise arguments and the reviewer did not participate to discussion.
With the other reviewers a consensus has been found for saying that the paper has merits and studies an important and interesting problem, however the reviewers agree that the paper needs a major revision, in particular related to the experiments, and that the current version is not ready for ICLR.
I follow the general recommendation and propose rejection.